# Seasonal variability of ocean circulation near the Dotson Ice Shelf, Antarctica

H. W. Yang[1,2], T.-W. Kim [1✉], Pierre Dutrieux [3,4], A. K. Wåhlin [5], Adrian Jenkins [6], H. K. Ha [7], C. S. Kim [8], K.-H. Cho [1], T. Park[1], S. H. Lee[1] & Y.-K. Cho[2]

Recent rapid thinning of West Antarctic ice shelves are believed to be caused by intrusions of warm deep water that induce basal melting and seaward meltwater export. This study uses data from three bottom-mounted mooring arrays to show seasonal variability and local forcing for the currents moving into and out of the Dotson ice shelf cavity. A southward flow of warm, salty water had maximum current velocities along the eastern channel slope, while northward outflows of freshened ice shelf meltwater spread at intermediate depth above the western slope. The inflow correlated with the local ocean surface stress curl. At the western slope, meltwater outflows followed the warm influx along the eastern slope with a ~2–3 month delay. Ocean circulation near Dotson Ice Shelf, affected by sea ice distribution and wind, appears to significantly control the inflow of warm water and subsequent ice shelf melting on seasonal time-scales.

[1] Korea Polar Research Institute, Incheon 21990, South Korea. [2] School of Earth and Environmental Sciences/Research Institute of Oceanography, Seoul National University, Seoul 08826, South Korea. [3] British Antarctic Survey, Natural Environment Research Council, Cambridge CB3 0ET, UK. [4] Lamont-Doherty Earth Observatory of Columbia University, Palisades, NY 10964, USA. [5] Department of Marine Sciences, University of Gothenburg, Gothenburg, Sweden. [6] Department of Geography and Environmental Sciences, Northumbria University, Newcastle upon Tyne NE1 8QH, UK. [7] Department of Ocean Sciences, Inha University, Incheon 22212, South Korea. [8] National Institute of Fisheries Science, Busan 46083, South Korea. ✉email: twkim@kopri.re.kr

Glacier flow in West Antarctica has been increasing[1–5], which can impact global sea-level rise[6,7]. The cause for the recent flow increase is believed to be the thinning of the buttressing ice shelves[1–3,8]. At the Amundsen Sea Embayment (ASE), ice from the West Antarctic Ice Sheet (WAIS) is drained into the ocean through the Pine Island, Thwaites, Haynes, Smith, Pope, and Kohler glaciers. These glaciers had an ice flux of $334 \pm 15$ Gt yr$^{-1}$ in 2013[9], and they have potential to impact sea level rise globally should this flux change significantly. In the 1980s, Pope, Smith, and Kohler Glaciers, located in the western ASE, drained ice about 15% of the total ice mass loss from the ASE, but since 2013 their rate have increased and they now contribute about 23%. In the ASE, warm and salty circumpolar deep water (CDW) can intrude from the deep ocean across the continental shelf under the influence of wind and Earth's rotation[10]. This happens in submarine troughs where the seabed is sufficiently deep (>500 m), and in the southern end of these troughs a modified version of CDW (slightly colder and fresher water compared with CDW; mCDW) can access deep-draft ice which accelerates the ice shelf melt and the glacier mass loss[11–17].

The oceanic heat transport in the AS is more significant than the Ross and Weddell Seas adjacent to the giant ice shelves in Antarctica. In the Ross and Weddell Seas, where large cyclonic polar gyres are located north of the continental shelf, only a tiny amount of CDW cooled within the gyre intrusion on the continental shelf[2,18]. Understanding seawater circulation near the ice shelf is essential for determining how changes in oceanic heat transport affect basal melting of ice shelves[19]. Recently, it has been found that the surface water flowing into the cavity influences Ross and Filcher-Ronne ice shelves melting and their seasonality[20,21]. Long-term mooring observation in front of Pine Island Glacier (PIG) have shown that the variation in sea surface heat flux can affect the variability in CDW volume and seawater circulation pattern[22]. Moreover, the seasonal expansion and contraction of polynya and sea ice melting and formation can affect the seawater density structure, affecting the seawater circulation.

The Dotson Ice Shelf (DIS), located southwest in the ASE, buttresses the Kohler and Smith Glaciers and has thinned by 2.6 m yr$^{-1}$ between 1994 and 2012, which is 30% faster than the average thickness change in the AS sector (1.94 m yr$^{-1}$)[5]. Observations near DIS has been conducted semi-regularly in summertime since 1994 and demonstrate a substantial inter-annual variation connected with the weather systems near the ASE[15,19]. Continuous time series, based on moored instrumentation, are rarer. Although there is a clear seasonal variation of the mCDW intrusions further north in the Getz–Dotson trough[10,23], previous records at the ice shelf front[15,24,25] have been too short to determine any seasonality there. The seasonal variation of atmospheric forcing into the ocean caused by the seasonality of the sea ice distribution will affect the ocean circulation near the ice shelf[26]. Therefore, confirming the seasonality of the mCDW circulation in front of the ice shelf and identifying the causes is essential for quantifying the influx of oceanic heat into the ice shelf cavity and understanding the thinning process of the ice shelf. In this study, we show that there is a clear seasonal variability of mCDW water flowing into the DIS and how it propagates into the cavity beneath the ice shelf. In addition, temporal and spatial variations in ocean circulation and heat transport to and from the sub-DIS cavity and look into associated forcing mechanisms are investigated.

## Results

**Observation.** In January 2014, three bottom-moored arrays were deployed from the South Korean icebreaker RVIB Araon along the DIS calving front to measure the variability of temperature, salinity, and currents. These moorings were located near the center of the Dotson–Getz trough (DGT) and on its eastern and western slopes, 2–3 km away from the calving front (K4, K3, and K5 in Fig. 1a). The location of the eastern (K4) and western (K5) moorings were chosen to measure mCDW inflows and meltwater outflows, respectively. All but one acoustic Doppler current profiler (ADCP) recorded successfully for 2 years, while the ADCP measuring near-bottom currents on the eastern mooring provided a 1-year record (Supplementary Table 1, Supplementary Fig. 1).

**Seasonal variation of mCDW at the eastern and western flank.** South-westward currents predominated near the bottom on the eastern side throughout the record, with an average current speed near 17 cm s$^{-1}$ at 680 m (Fig. 1a). Away from the seabed, the current was somewhat weaker (near 5 cm s$^{-1}$ at 400 m) and veered westward. Near the western flank, weak south-eastward flows (around 1 cm s$^{-1}$) were measured at 700 m (Fig. 1b). The current direction gradually turned eastward at shallower levels, with an average current speed reaching 1.2 cm s$^{-1}$ at 420 m (Fig. 1a). Warm and salty mCDW (2 °C higher than the seawater freezing point at the surface) was found flowing towards DIS at the eastern flank below 700 m (Fig. 1c, d), with average temperature and salinity for two years at 747 m near the seabed being 0.3 °C and 34.5 PSU, respectively. In contrast, the outflows of relatively cold and fresh water (−1 °C, 34.2 PSU) were found above 500 m near the western flank (Fig. 1b, e).

The predominantly south-westward flow on the eastern flank (Fig. 2a, b) displayed large, depth-independent seasonal and intra-seasonal variability. Near the bottom (680 m), the southward (towards the ice shelf cavity) flow reached a maximum of 20 cm s$^{-1}$ in January 2014 and a minimum of 12 cm s$^{-1}$ in May. The southward component decreased gradually at mid-depth, but its seasonal and intra-seasonal variability remained the same in the whole water column. At the 400 m depth, the observed maximum southward component was 9 cm s$^{-1}$ in January 2016, and a northward component of 2 cm s$^{-1}$ was measured in April 2015. The vertical shear of the meridional current component was largest in the warm layer near the bottom, and it had comparatively small temporal variations through the measured period. This suggests, that the seasonality of the southward component is a barotropic process, and that the baroclinic component (associated with the vertical shear) does not significantly affect the seasonal variability of the southward flow.

In summer, the strong barotropic southward flows drive an increase of salinity in front of the DIS (Supplementary Fig. 2) because the salinity is relatively low in front of the DIS due to the strong down-welling[26]. Salinity at mid-depths (above 500 m) varied seasonally near the eastern and central front, as shown by the 34.2 PSU isohaline, rising from ~450 m in early spring to ~300 m in early summer and deepening 450–500 m for the rest of the year. The seawater temperature does not differ significantly between the eastern slope and the center, similar to the salinity. Thus, the baroclinic effect by density gradient seems insignificant in the mid-depth layer. This spatial distribution of salinity suggests that the vertical shear of the southward flow (Fig. 2b) is less prominent in the middle layer (400–560 m) compared to the lower layer (below 560 m). Nearer the seabed there was no distinct seasonal variability of the salinity, but a noticeable eastward gradient increased in association with intrusions of mCDW along the eastern flank. The salinity at 750 m was above 34.5 PSU on the eastern side, 34.4 PSU in the center, and 34.35–34.4 PSU on the western side. Such a salinity gradient is associated with an increase of southward current shear near the bottom.

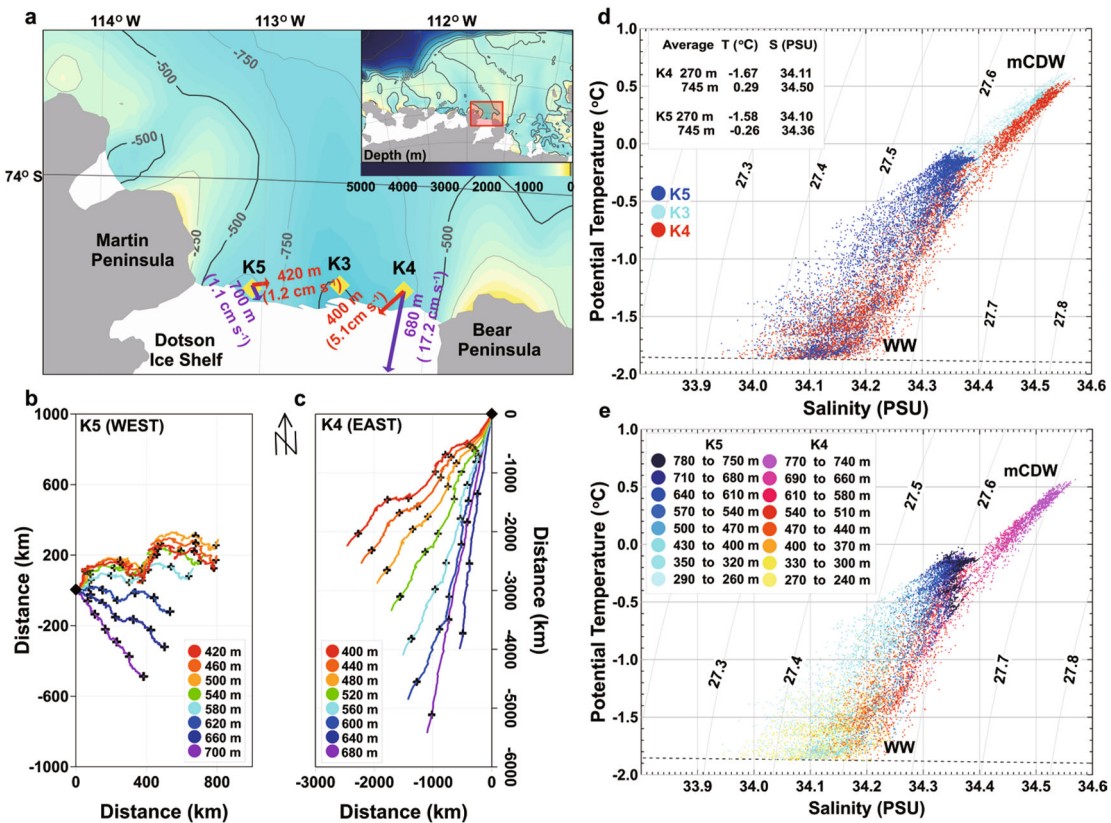

**Fig. 1 Study area, bathymetry, station locations, progressive vector and temperature–salinity diagrams at mooring sites. a** Bathymetry, coastline[66], and stations on the Amundsen Sea continental shelf (location shown in red rectangle). Yellow diamonds show moorings K3–K5 near the Dotson Ice Shelf. Red and purple arrows are average current at the top layer (K4 = 400 m, K5 = 420 m) and the bottom layer (K4 = 680 m, K5 = 700 m). **b, c** Progressive vector diagrams of velocities at K5 from 420–700 m depth and K4 from 400–680 m depth during 2014 and 2015. Color-coded dots denoted depth and marked the black cross on the diagrams every 3 months interval. Black diamonds are the starting point. **d** Temperature–salinity at K4 (red), K3 (cyan), and K5 (blue). Average potential temperature and salinity near the top (270 m) and bottom layer (745 m) are highlighted. **e** Temperature–salinity color-coded by the depth at K4 and K5.

**Effect of ocean surface forcing**. North of the DIS, the Amundsen Sea polynya (ASP) repeats seasonal expansion and contraction, which can affect seawater circulation and spatial distribution of mCDW by causing changes in ocean surface density during sea ice formation and melting. Previous study from PIG[22] suggests that local positive (i.e., from ocean to air) buoyancy fluxes from sea ice formation and/or atmospheric cooling (less buoyant at the sea surface) creates deep convection, leading to a downward descent of the thermocline and thinning of the mCDW layer at the bottom, while negative buoyancy fluxes (i.e., downward and more buoyant at the sea surface) due to surface heating and sea ice melting leads to an upward movement of the thermocline and a increase mCDW volume. In order to investigate the effect of sea ice fluctuations on the change of mCDW and its circulation, local surface buoyancy fluxes were calculated using the heat- and freshwater fluxes from the data-assimilating Southern Ocean State Estimate (SOSE)[27]. These surface buoyancy fluxes were then compared to the depth of isohalines and meridional velocities (Supplementary Fig. 3a and see the section "Methods"). Both local buoyancy flux and meridional velocity demonstrate a seasonal variation that decreased in summer and increased in winter. The buoyancy flux, which affects the density structure of the water column, modulates the baroclinic component of the meridional velocity. However, the variation range of the baroclinic component (depth-averaged removed) in meridional velocity was 5.67 cm s$^{-1}$ at 680 m depth, which was the largest in the entire layer, but almost half of the barotropic component of 11.52 cm s$^{-1}$.

Therefore, although both buoyancy flux and meridional velocity are influenced by atmospheric variability and vary seasonally, the effect of local buoyancy fluxes on mCDW variability and circulation are comparatively weak in the present data set.

At the outer edges of the ASP, there is a boundary between open water and (more or less) fast sea ice. When a homogeneous wind field blows over an opening in the fast ice, spatial stress gradients (ocean surface stress curl, OSSC, Fig. 2c) are created near the edges, resulting in divergence or convergence of the wind-driven surface (Ekman) transport[23]. These can induce barotropic currents. In other to estimate the effect of the barotropic component on the variability of the southward current near the eastern side of DIS front, the OSSC was calculated from sea ice motion and wind using the ice–ocean drag coefficient (see the "Methods" section and Supplementary Fig. 4) and compared to the mooring data. The variability of OSSC on the eastern flank was relatively high compared to the western flank. Cross-spectral analysis of the OSSC at the eastern flank (74°S, 112.25°W) and the vertically averaged southward velocity at mooring K4 showed a statistically significant coherence around 80-day frequency with other shorter frequencies (e.g., 31-day and 22-day) (Supplementary Fig. 5). In addition, 20-day low-pass-filtered southward velocity had a statistically significant correlation ($r = 0.47$) with OSSC at 28-day lag (Fig. 2d, e), approximately a quarter of a combines of a dominant 80-day frequency and other shorter frequencies, suggesting that the accumulated OSSC forces variation in the barotropic meridional velocity. That is, positive OSSC along

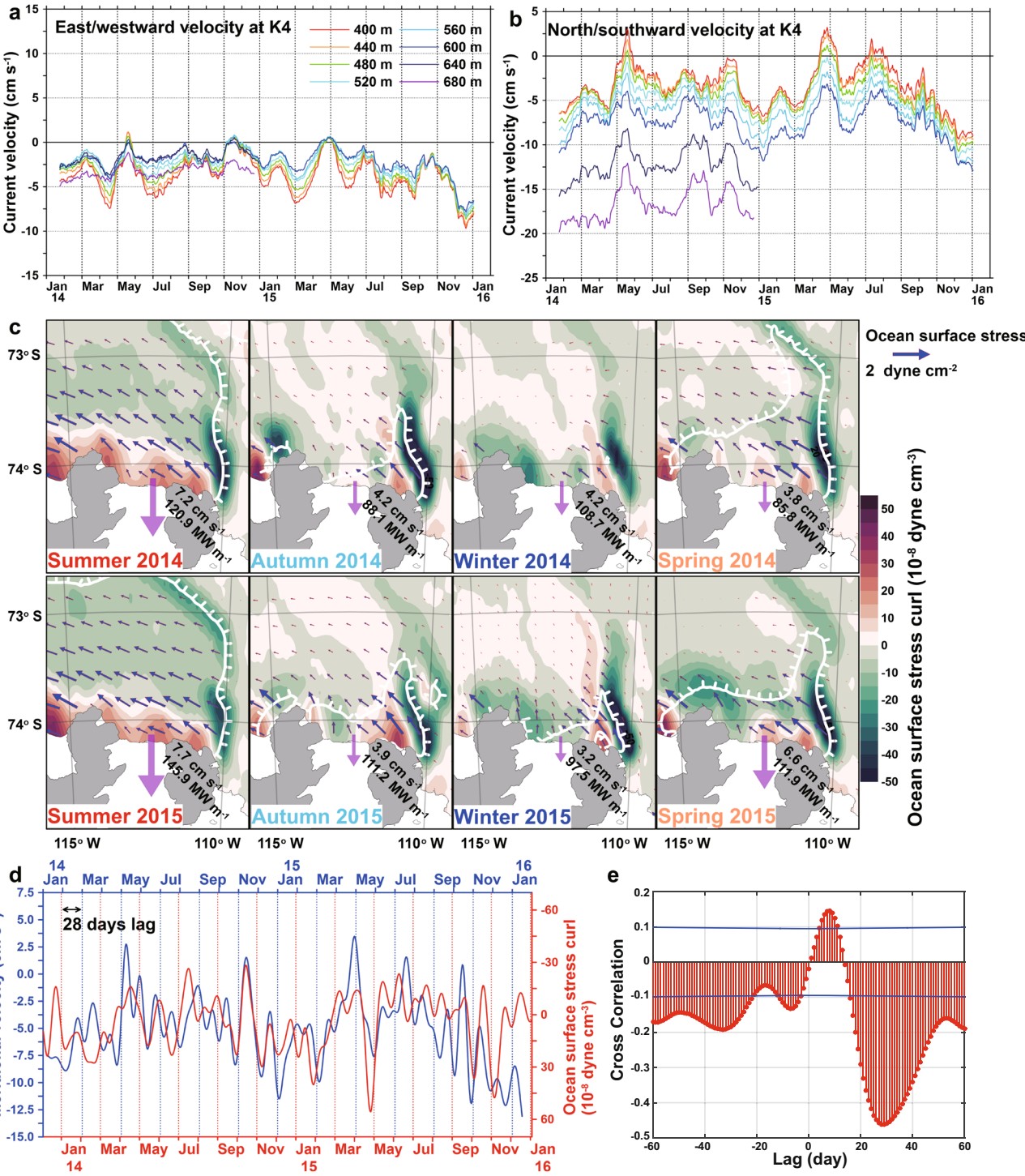

**Fig. 2 Time series of velocities and ocean surface stress curl. a, b** 31-day running average zonal and meridional velocities at K4 (East flank). Positive values indicate east and northward currents. **c** Horizontal distribution of ocean surface stress curl (color), ocean surface stress (arrow) and sea ice concentration (SIC) in 2014 and 2015. Outside of the white line indicates where SIC is over 50%. A positive ocean surface stress curl (OSSC) means high sea surface height. The seasonal average southward velocity and heat transport are shown on the map, and the purple arrows indicate the magnitude of southward heat transport. **d** 20-day low-pass filtered OSSC at 74°S, 112.25°W (red) and vertical mean (400–600 m) meridional velocity (blue). Axis of the OSSC was reversed. **e** Cross-correlation between OSSC and velocity with 20-day low-pass filter and 99% confidence interval (blue line).

the eastern slope of the DIS increases the cumulative OSSC, increasing the horizontal gradient of the sea surface elevation and accelerating the barotropic southward flow. When the positive OSSC turns negative, the barotropic southward flow is maximum and then decelerates. We interpret this statistically significant relationship as an indication that higher OSSC over the eastern

flank creates a zonal barotropic pressure gradient driving a southward velocity. During summer, the OSSC increases due to strong south-easterly winds over the open sea and during winter it decreases when the wind stress and currents are weakened by the sea ice cover (Fig. 2c). The summertime increase of the OSSC leads to a strengthened southward mCDW flow (Supplementary Fig. 6a)

and enhanced heat transport to the ice shelf. Conversely, both southward flow and heat transport decrease in winter. The relatively high OSSC near the DIS can also induce a barotropic pressure gradient in the meridional direction, which may similarly drive the strongly barotropic westward Antarctic coastal current at mid-depth (Fig. 2a). The westward current component decreases with depth due to the eastward baroclinic shear caused by the local down-welling associated with high OSSC near the ice shelf front.

On the western flank of the ice shelf front, the seasonal variation, most prominent in mid-depth, had a range about half that of the eastern side bottom (Fig. 3a, b). Near the bottom, near-constant weak south-eastward currents were observed during the entire record. The meridional current component shifted between northward in fall and southward in spring (Fig. 3b) and the current velocities were generally smaller compared to the eastern flank. In both 2014 and 2015, the highest northward velocities were 2.9 and 2.5 cm s$^{-1}$ in April and the highest southward velocity was 2.5 cm s$^{-1}$ in early October. The eastward current component had a maximum in winter, 4.2 and 4.9 cm s$^{-1}$ at 420 m depth in July (Fig. 3a), and was almost constant close to 2 cm s$^{-1}$ in summer. The occurrence of eastern flow in the winter coincides with a decrease of OSSC at the western side of DIS front (Fig. 3c), and an associated local drop in sea level height, intensifying the eastward barotropic current in the entire water column. However, the upwelling accompanied by negative OSSC generates a westward baroclinic current that countervails the eastward barotropic current at a deeper in the water column.

**Meltwater outflow.** During all seasons, there was a weak southward flow present near the bottom throughout the record at the western side of DIS front (Fig. 3b), presumably driven by the cross-front salinity distribution which increased toward the eastern side of DIS due to the inflow of mCDW. In the middle layer, the northward current component had a seasonal maximum in autumn, both years, coinciding with maximums of the meltwater fraction (e.g. concentration of glacial meltwater in seawater calculated by Eqs. (3) and (4) in the "Methods" section) (Fig. 3b, d). These meltwater outflows occurred mainly in the layer between 272–540 m and can be significant enough to influence the density structure in the water column and thereby the local circulation[26,28]. Since the mooring did not cover the upper water column, it is believed that much of the meltwater outflow remained undetected (Supplementary Fig. 7). However, the signal is sufficiently strong to identify the seasonal variation of the outflow. We observe a seasonal variation of the meltwater fraction (Fig. 3d, see the "Methods" section), with a maximum exceeding 1% in April for 275 m depth in both years. This value is similar to PIG (~1.5% at 100–500 m)[29] and Shirase Glacier Tongue in East Antarctica (near 0.8%)[30]. The seasonal phase of the maximum meltwater fraction is delayed gradually toward the bottom, with the maximum at 575 m (0.45%) appearing in July. In contrast, the minimum meltwater fraction appeared in early October/September both years for the whole water column.

A decrease of vertically integrated density along the western side of the ice front by a discharge of meltwater during autumn leads to local sea surface elevation rise and an increase in the northward barotropic current. There is a strong correlation ($r = 0.9$, with a 17-day lag) between the meridional current velocity and the upper layer meltwater fraction (Fig. 3e and Supplementary Fig. 8). Thus, the variability of meridional current at the western side of DIS may be influenced by meltwater discharge. This is in contrast to the eastern side, where it is primarily OSSC that changes the velocity. Deeper in the water column, the freshening by an inflow of meltwater in the water column reduces the density in the western front. The density

gradient increasing to the east causes an increase in the southward baroclinic current countervailing the northward barotropic current.

During summer and autumn, the input of glacier meltwater in the Dotson trough strengthens athe stratification and prevents vertical convection of dense water[31]. However, the decrease of meltwater discharge in the winter and spring would weaken the stratification and enhances the deep convection of dense winter water. An upper water column homogenized by vertical mixing can provide an opportunity for the winter water to descend to the middle layer. This strengthening of vertical convection and the Antarctic Coastal Current[26,29] to the westward can rapidly reduce the upper layer meltwater fraction (Fig. 3d).

**Relation between heat transport and meltwater flux.** In order to evaluate the effect of warm water inflow and its seasonality on the basal melting, heat transport toward the DIS near its eastern front was estimated as a function of heat content and southward current velocity for the two years (see the "Methods" section and Fig. 4). The heat transport varied between 51 and 182 MW m$^{-1}$, with summertime values more than three-fold those of winter due to the strengthened southward velocities and higher seawater temperature near the bottom. Although the peak of heat transport in winter was smaller than that in summer, it was conspicuous (July 2014 and June 2015, Fig. 4b) compared to the small peaks in spring and autumn. Such two winter peaks mainly depend on the strengthening of the southward flow and are related to the increase of OSSC due to the strengthening of the wind (Supplementary Fig. 9).

Although it is expected that the seasonal increase in heat transport has implications for the basal melt of the ice shelf, recent results indicate that the barotropic current component, which is a strong contributor to the seasonal variability, can be at least partially blocked at the ice front[32]. Nevertheless, the seasonal variability of heat transport in front of the ice shelf will be propagated to under the ice shelf because the barotropic southward flows affect the variability of isohalines and isotherms, and that signal is expected to propagate in the cavity (Supplementary Fig. 3). The oceanic heat transport along the eastern slope melts the DIS base, and a fresh meltwater mixture returns to the open sea along the western slope. Meltwater fluxes on the west side seemingly follow heat transports on the east side with a 74-day lag (correlation of $r = 0.50$ on a 99% significance level, Fig. 4b, and Supplementary Fig. 10). Such a significant correlation indicated that the heat transport to the ice cavity affected by the OSSC variability at the eastern slope causes the seasonal variation of meltwater discharge and northward flow at the western slope. The observed delay between warm inflow and meltwater outflow agrees qualitatively well with the two months residence time inside the cavity observed in 2018[33]. Our new observation on the variability of seawater circulation in front of ice shelves and the link between heat inflow to the ice cavity and meltwater discharge complements the previously revealed mechanism by Jenkins et al. (2018)[34] that heat content variability in front of DIS can affect the heat transport into the DIS cavity and ice shelf melt.

## Discussion

Using two years of mooring data near the calving front of Dotson ice shelf, we identified a substantial inflow of warm and salty water near the seabed at the eastern flank of the deep trough that leads into the ice shelf cavity. We also identified high concentrations of meltwater on the opposite (western) flank. These findings have been observed in other systems that receive warm salty water through deep troughs in the continental shelf, such as Getz ice shelf[32,35], Pine Island Glacier[19], Amery ice shelf[36], as well

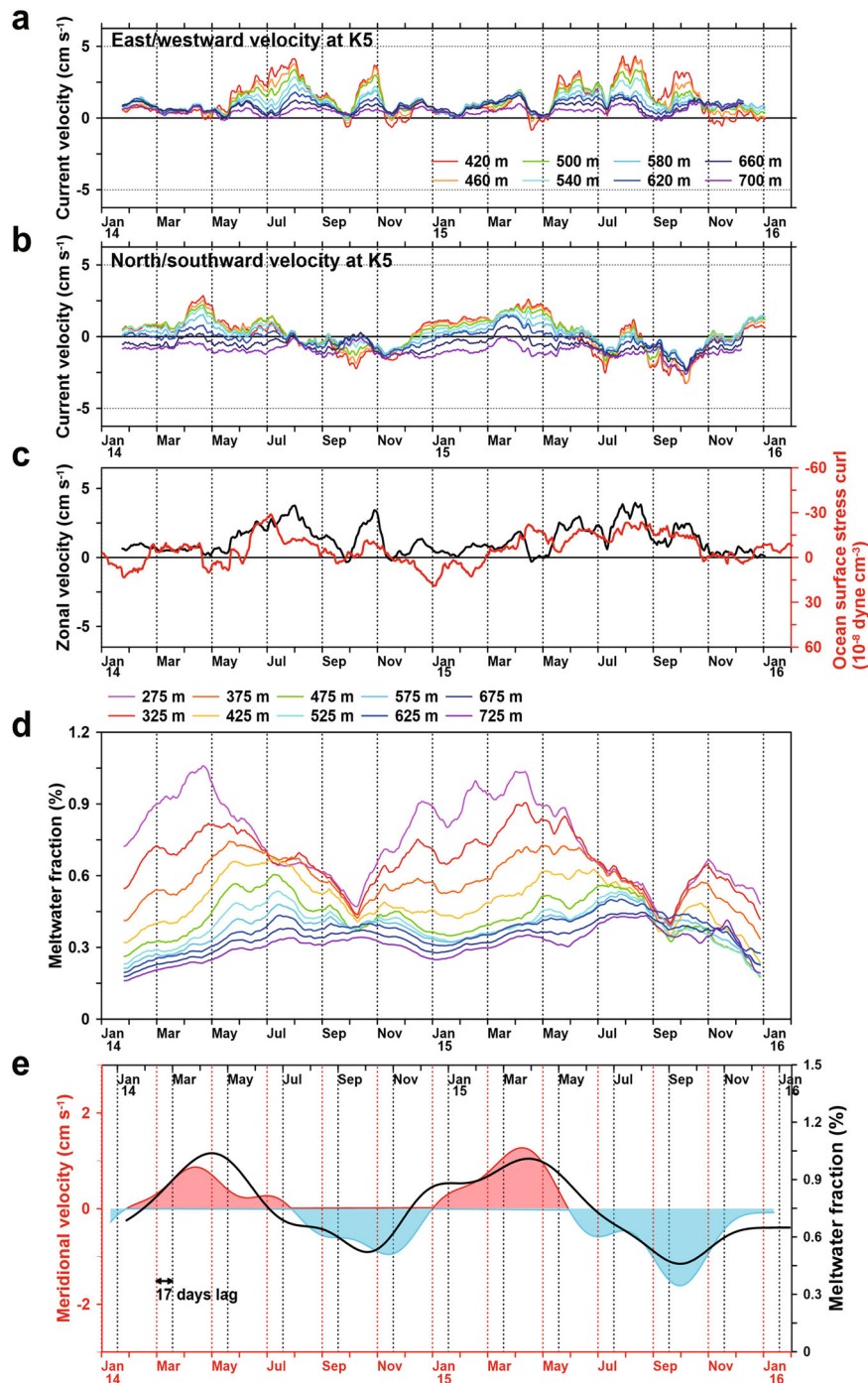

**Fig. 3 Time series of velocities and meltwater fraction at K5. a** and **b** 31-day running average zonal and meridional velocities at K5 (West flank). **c** 31-day running average zonal velocity (420 m, black line) and mean ocean surface stress curl (OSSC) (74°S, 112.75–113.25°W, red line). $r = 0.57$, confidence level = 99%. **d** 31-day running average meltwater fraction at 50 m intervals from 275 to 725 m (thin color-coded lines). **e** Velocity (vertical mean) and meltwater fraction (black line) at 275 m with 90-day low-pass filter ($r = 0.9$, with a 17-day lag). Red and blue shades indicate northward and southward flow, respectively. The upper x-axis is back-shifted by 17-day.

as Dotson trough further north than the presently studied system[37]. A similar circulation pattern has also been noted in comparatively 'cold' shelf systems such as Ross ice shelf[20] and Filchner-Ronne ice shelf[38], although it does not substantially influence the ice shelf melt processes there. It was also shown that high meltwater concentrations correlate with intermittent outflows from the western side of the cavity, which is to be expected but is rarely observed due to the challenging environments for data collection near the calving fronts of these large glaciers.

Both the warm inflow in the east and the meltwater outflow in the west had a clear seasonal variation. Inflow velocity and heat content in the east peaked in summer, with a nearly three-fold increase of the summertime heat transport toward the cavity. There was also an autumn maximum in both meltwater content and outflow velocity in the western side of the ice shelf front, delayed by about 2 months from the inflow peak in the east. The substantial seasonal variability of heat transport and meltwater discharge near the ice shelf calving front, that was observed,

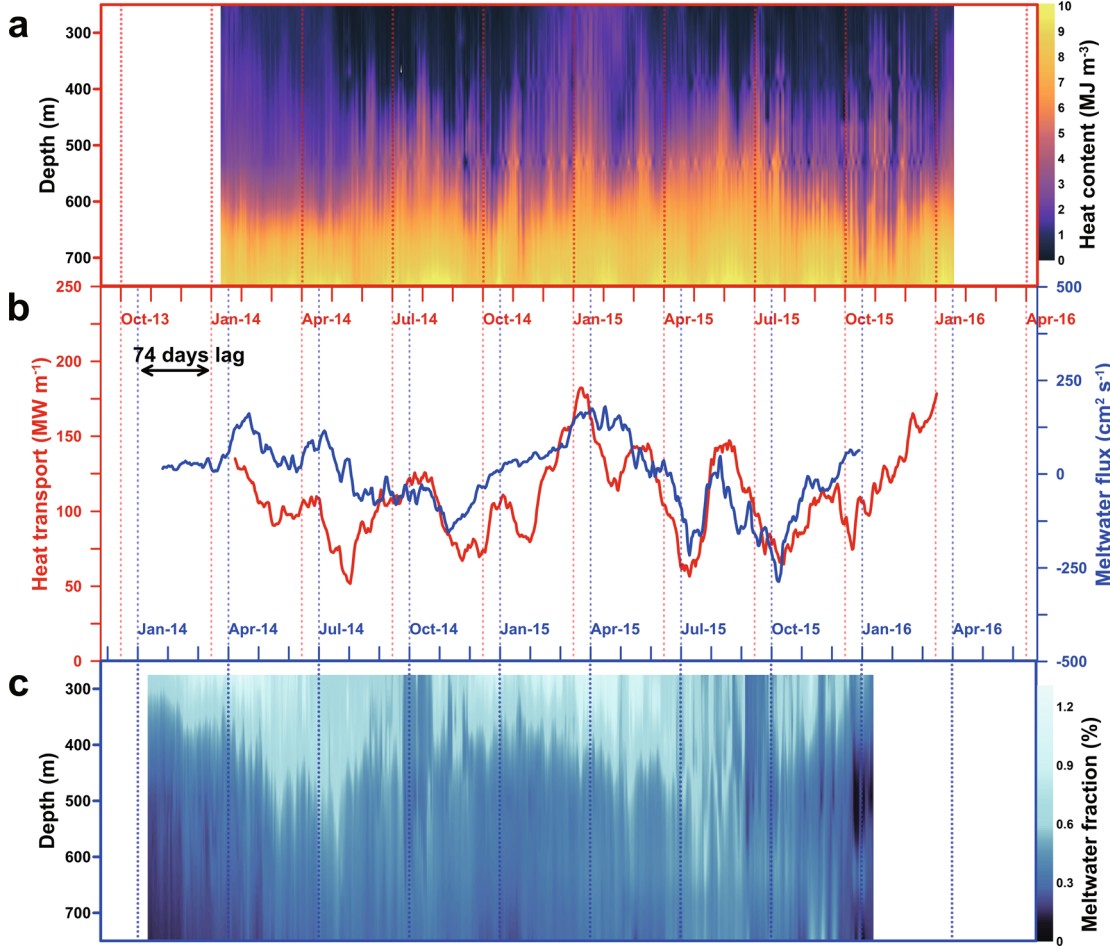

**Fig. 4 Time series of heat content and heat transport at K4 and meltwater fraction and meltwater flux at K5. a** Vertical and temporal distribution of calculated heat content from 250 to 750 m at K4. **b** 31-day running average vertical mean meltwater flux(blue) between 275 and 575 m at mooring K5 and estimated heat transport (red) at mooring K4. The upper *x*-axis for heat transport and content has been back-shifted 74-day. **c** Vertical and temporal distribution of meltwater fraction at mooring K5.

suggests that previously evaluated heat- and meltwater transport[15,24,25], based on summer observation, may have been overestimated. Based on the present data set the average heat transport in summer was 141 MW m$^{-1}$, 1.27 times greater than the annual average of 111 MW m$^{-1}$, a number that can be used to scale summertime observations from this region. The seasonal average meltwater flux to the north was at a maximum of 89.2 cm$^2$ s$^{-1}$ in autumn and 25.7 cm$^2$ s$^{-1}$ in summer. In contrast, meltwater flux showed negative in winter and spring due to the dominant southward flows, and the annual average was close to zero. This seasonality implies that additional observations are needed to determine how much of the inter-annual variability of heat and meltwater transport seasonality propagates into the ice shelf cavity. The effects of the strong seasonality also have implications for timing of snapshots used to calculate ice shelf thickness evolution and points to the importance of high-resolution remote sensing products when calculating ice shelf mass balance.

The seasonal variation observed at DIS is in contrast to seasonal variation in the deep troughs leading to Pine Island Glacier[22] and to Getz Glacier[32,35,39], where a wintertime maximum in heat content and velocity has been observed. In contrast to the situation at Pine Island Glacier calving front[22], where the main cause for vertical migration of the thermocline is local buoyancy flux, short-term variability of heat transport was here caused by local changing stress at the ocean surface due to wind

and sea ice conditions. These appear to be similar to the intermittent reductions in heat transport that have been observed at the Getz ice shelf cavity[35], except that the forcing in the present locality is mainly local while the Getz ice shelf appears to be predominantly remotely forced and transmitted via topographic waves to the cavity opening.

The driving mechanism behind seasonality of the heat transport into the DIS cavity is interpreted to be caused by local OSSC, induced by changes in sea ice and wind distribution, in turn linked to changes of the polynya area and sea ice distribution[40]. Previous studies have shown that the wind field and atmospheric circulation affects not only the seawater circulation at the ice shelf front but also the variability of warm water flowing into the DGT near the center of the continental shelf[10,11,23,41]. Also, local wind forcing in front of PIG has been attributed a significant modulator for the short-term (less than one month) variability of the upper thermocline depth and basal melting there[42]. Local atmospheric forcing, such as wind, is the most crucial factor in determining the variability of the seawater circulation and oceanic heat transport in the AS. Meanwhile, atmospheric circulation in the AS is much more complex and diverse than in any other sea in Antarctica due to the variability[43,44] (i.e., seasonal migration of longitudinal location) in the Amundsen Sea Low (ASL)[45]. In addition, this variability in ASL was influenced by the Southern Annular Mode (SAM) and El Niño-Southern Oscillation (ENSO) phases in a long-term timescale[46].

Through long-term observations in front of DIS for two years, we found that heat was delivered to the DIS cavity along the eastern flank of the DGT, and glacial meltwater was emitted along the western flank with a time lag of two months. Furthermore, the seasonality of heat transport and meltwater discharge was confirmed due to the variability of OSSC induced by wind and sea ice distribution. The circulation pattern of mCDW in the front of the ice shelf and its seasonal variability by atmospheric conditions demonstrated here allow for quantitative evaluation of the effects of short-term variability in the atmosphere on ocean circulation. This finding implies that the ocean circulation effect by local meteorological conditions in front of the ice shelf plays an essential role in regulating mCDW inflow and the basal melting of the ice shelf. Therefore, understanding the long-term variability of the atmosphere and the subsequent response of the ocean circulation are likely essential for determining the long-term melting trend of the WAIS. These results suggest that further studies are needed, e.g., ice-ocean coupled numerical model study considering the ice shelf.

## Methods

**Data collection.** To monitor the temporal and spatial variation of mCDW and its effect on the rapid melting of glaciers in the Amundsen Shelf, we conducted extensive oceanographic surveys 4 times from 2010 to 2016. The temporal variability and properties of mCDW and its circulation in front of the DIS were obtained from the K3–K5 moorings deployed on January 8–9 2014, and recovered on January 19–20, 2016. At each station, the current velocities were observed from dual ADCPs with upward-looking 150 kHz (K4 and K5) or 75 kHz (K3) and downward-looking 300 kHz (K3–K5) RD Instruments (RDI) (Supplementary Table 1). All ADCPs were configured in a narrowband mode for optimal range. The upward-looking 150 and 75 kHz ADCPs used 8 m bins and 15 min ensembles of 25 and 40 pings, respectively. The downward-looking 300 kHz ADCP used 4 m bins and 15 min ensembles of 20 pings. Unfortunately, the latter recorded velocities for only 371 and 713 days from January 2014 at K4 and K5, respectively, but the other ADCPs recorded continuously during the entire observation period (Supplementary Fig. 1). The observed velocities were processed using WinADCP® software and the tidal signal was removed using the t_tide toolbox[47].

In addition, the moorings contained 8 (K4 and K5) or 12 (K3) Sea-Bird Electronics (SBE) 37-SM or 37-SMP-ODO MicroCAT sensors (Supplementary Table 1) to observe conductivity as accuracy of 0.0003 S m$^{-1}$, temperature (accuracy of 0.002 K), and dissolved oxygen (accuracy of 3 μmol kg$^{-1}$).

**Heat transport calculations.** The heat content (HC, J m$^{-3}$) at each mooring was calculated relative to the freezing point:

$$HC = \rho_0 C_p (T - T_f) \tag{1}$$

where $\rho_0$ is the reference seawater density (1027 kg m$^{-3}$); $C_p$ is the specific heat of seawater (3986 J K$^{-1}$ kg$^{-1}$); and $T$ and $T_f$ are seawater temperature and seawater freezing point at the surface based on the salinity, respectively.

To calculate heat transport during the entire observation period from 400 to 680 m, velocity below 600 m depth since Jan. 2015 at K4 was estimated from the relationship between the meridional velocity of the 600 m layer and data every 20 m (620, 640, 660, 680 m) from January 2014 to January 2015. In this calculation process, it was assumed that the shear between each layer of the baroclinic component of southward velocity was always constant. The linear regression showed a high coefficient of determination from 0.93 to 0.64 (Supplementary Fig. 11). Heat transport in the 400–680 m layer was calculated from estimated meridional velocity and heat content:

$$Q = \int_{680}^{400} \rho_0 C_p (T - T_f) \times V \, dz \tag{2}$$

**Meltwater fraction calculations.** The outflow water from DIS was considered a mixture of winter water (WW), mCDW, and ice shelf meltwater based on the assumption that the ice-seawater system was closed. This relationship was expressed by water mass ($v$) and properties ($\chi^1, \chi^2$)[48]:

$$v_{cdw} + v_{ww} + v_{melt} = 1 \tag{3.1}$$

$$v_{cdw}\chi^1_{cdw} + v_{ww}\chi^1_{ww} + v_{melt}\chi^1_{melt} = \chi^1_{obs} \tag{3.2}$$

$$v_{cdw}\chi^2_{cdw} + v_{ww}\chi^2_{ww} + v_{melt}\chi^2_{melt} = \chi^2_{obs} \tag{3.3}$$

Following the conservation equation for the above equation, the meltwater fraction ($\varphi$) was then calculated from observed in situ temperature and salinity and the end-members of ice and water mass[24,34,49,50]:

$$\psi^{T,S}_{melt} = (\chi^T_{melt} - \chi^T_{CDW}) - (\chi^S_{melt} - \chi^S_{CDW})\left(\frac{\chi^T_{WW} - \chi^T_{CDW}}{\chi^S_{WW} - \chi^S_{CDW}}\right)$$

$$\psi^{T,S}_{mix} = (\chi^T_{obs} - \chi^T_{CDW}) - (\chi^S_{obs} - \chi^S_{CDW})\left(\frac{\chi^T_{WW} - \chi^T_{CDW}}{\chi^S_{WW} - \chi^S_{CDW}}\right) \tag{4}$$

$$\varphi = \frac{\psi^{T,S}_{mix}}{\psi^{T,S}_{melt}}$$

where $\chi^T$ and $\chi^S$ represent potential temperature and salinity and subscripts melt, CDW, WW, and obs indicate the end-members of ice, CDW, Winter water, and in situ observations at K5. In previous studies, meltwater fraction calculation and end-member selection have used summer observation data[24,34,48–50]. In this study, an end-member applicable over the entire season was required to calculate the meltwater fraction using mooring data. However, winter water is generated as a result of brine rejection and convection by sea surface cooling and remains until summer above the mCDW layer. Thus, to detect this 'pure' winter water, we selected end-member estimated from summer observation already presented. We used the two-year average (2014 and 2016) end member for CDW ($T \sim 0.013$ °C, $S \sim 34.46$ PSU), WW ($T \sim -1.88$ °C, $S \sim 34.24$ PSU) and ice ($T \sim -95$ °C, $S \sim 0$ PSU) obtained from previous studies in the AS[34].

**Buoyancy flux calculation.** Buoyancy flux is caused by atmospheric heating and formation and melting of sea ice. Buoyancy flux was calculated as

$$Bf = -g\alpha\frac{Q_{HF}}{\rho_w C_p} + g\beta Q_{FW} S_0 \tag{5}$$

where $g$ is acceleration of gravity (9.8 m s$^{-2}$) and $\alpha$ at the sea surface (35 PSU, 0 °C) is the thermal expansion coefficient (5.1 × 10$^{-5}$ K$^{-1}$)[51]. $Q_{HF}$ is net air–sea heat flux (W m$^{-2}$) and positive means that the ocean gains heat. $\rho_w$ and $C_p$ are water density and specific heat of seawater (4190 J kg$^{-1}$ K$^{-1}$), respectively. $\beta$ (at the sea surface, 35 PSU, 0 °C) is the saline contraction coefficient (7.9 × 10$^{-4}$)[51]. $Q_{FW}$ is the freshwater flux (m s$^{-1}$) and negative means that the ocean gains fresh water. $S_0$ is the ocean surface salinity. $Q_{HF}$ and $Q_{FW}$ are obtained from the data-assimilating Southern Ocean State Estimate (SOSE) for 1/6° from 2013 to 2018[27].

**Drag coefficient between the air, sea ice, and ocean.** The processes of momentum exchange between the atmosphere and ocean are complicated on the continental shelf around Antarctica because wind forcing will be delivered to the ocean through sea ice. Therefore, the atmospheric and oceanic drag coefficients are salient parameters for estimating the effect of wind forcing on the variability of seawater circulation. The drag coefficient is determined by the speed reduction and veering of sea ice. We obtained wind data from the Antarctic Mesoscale Prediction System (AMPS), which uses the Polar Weather Research and Forecasting (WRF) model, the mesoscale model specially adapted for polar regions[52] providing gridded wind data above 10 m at the sea surface with a horizontal resolution of 20 km (March 6, 2006 to October 31, 2008), 15 km (November 1, 2008 to December 31, 2012), and 10 km (January 1, 2013 to December 31, 2018) at 3 h intervals (Supplementary Fig. 4b). Sea ice concentration data with a horizontal resolution of 3.125 km were obtained from the Advanced Microwave Scanning Radiometer for Earth observing system (AMSR-E, January 1, 2006 to October 4, 2011), the Special Sensor Microwave Imager/Sounder (SSMIS, October 4, 2011 to July 1, 2012), and the Advanced Microwave Scanning Radiometer-2 (AMSR-2, July 2, 2012 to present)[53]. Although the horizontal resolution of SSMIS is relatively low compared with the other two datasets, we interpolated the sea ice concentration to the same grid spacing as the AMSR-E. For sea ice velocity, data provided by the Polar Pathfinder Daily 25 km EASE-Grid Sea Ice Motion Vectors Version 4 from 2006 to 2018 ware used[54].

The free drift sea ice motion using the Coriolis force, balance of wind, and water stress[55] then becomes:

$$\rho_a C_{D,ai}|W_{10}|W_{10} + R(\theta_w)\rho_w C_{D,io}|U_w - U_{ice}|(U_w - U_{ice}) + R\left(\frac{\pi}{2}\right)\rho_{ice}hf(U_w - U_{ice}) = 0 \tag{6}$$

where $C_{D,ai}$ is drag coefficient between the air and ice; $W_{10}$ is surface wind; $U_w$, and $U_{ice}$ are the ocean current and sea ice velocities, respectively; $\rho_a$, $\rho_w$, and $\rho_{ice}$ are the densities of air, water, and sea ice, respectively; $f$ is the Coriolis parameter; and $h$ is ice thickness. The rotation matrix $R$ as a function of the angle ($\theta$) between ($U_{ice} - U_w$) and $W_{10}$ is given by

$$R(\theta) = \begin{pmatrix} \cos\theta & -\sin\theta \\ \sin\theta & \cos\theta \end{pmatrix} \tag{7}$$

The $U_{ice}$ as sea ice velocity can be expressed as

$$U_{ice} = \alpha R(-\theta)W_{10} + U_w \tag{8}$$

where $\alpha$ is the wind factor. In an idealized steady ocean, the second and third terms in Eq. (6) can be rewritten as $-R(\theta_w)\rho_w C_{D,io}|U_{ice}|(U_{ice})$ and $-R(\pi/2)\rho_{ice}hfU_{ice}$, assuming that the current velocity is close to zero[56]. The ice–ocean drag coefficient ($C_{D,io}$) can be estimated, allowing sea ice motion and sea ice velocity in free drift to

be defined as

$$\alpha^4 + N_a^2 R_O^2 \alpha^2 - N_a^4 = 0 \tag{9}$$

where $N_a$ is the Nansen Number and $R_o$ is the ice's Rossby Number, given by

$$N_a = \sqrt{\frac{\rho_a C_{D,ai}}{\rho_w C_{D,io}}} \text{ and } R_o = \frac{\rho_{ice} H f}{\rho_w C_{D,io} N_a |W_{10}|} \tag{10}$$

The wind factor ($\alpha$) and angle ($\theta$) can be written as

$$\alpha = N_a \sqrt{\frac{\sqrt{R_o^4 + 4} - R_o^2}{2}} \tag{11}$$

$$\theta = \arctan\left(\frac{N_a R_o}{\alpha}\right) = \arctan\left(\frac{1}{\sqrt{\sqrt{\frac{1}{4} + \frac{1}{R_o^4}} - \frac{1}{2}}}\right) \tag{12}$$

The observation data can be calculated from each grid point as[57]

$$\alpha = \frac{\cos(\theta)\sum W_{10}^{x'} U_{ice'}^{x} + \sin(\theta)\sum W_{10}^{y'} U_{ice'}^{x} - \sin(\theta)\sum W_{10}^{x'} U_{ice'}^{y} + \cos(\theta)\sum W_{10}^{y'} U_{ice'}^{y}}{\sum W_{10}^{x'^2} + \sum W_{10}^{y'^2}} \tag{13}$$

$$\theta = \arctan\left(\frac{\sum W_{10}^{x'} U_{ice'}^{y} - \sum W_{10}^{y'} U_{ice'}^{x}}{\sum W_{10}^{x'} U_{ice'}^{x} - \sum W_{10}^{y'} U_{ice'}^{y}}\right) \tag{14}$$

Finally, using the functions above for $\theta$ and $\alpha$, the expressions for $N_a$ and $R_o$ become:

$$N_a = \frac{\alpha}{\sqrt{\frac{\sqrt{\frac{1}{\left(\frac{1}{\tan^2\theta} + \frac{1}{2}\right)^2} - \frac{1}{4}} + 4 - \frac{1}{\sqrt{\left(\frac{1}{\tan^2\theta} + \frac{1}{2}\right)^2} - \frac{1}{4}}}{2}}} \tag{15}$$

$$R_o = \frac{1}{\sqrt[4]{\left(\frac{1}{\tan^2\theta} + \frac{1}{2}\right)^2 - \frac{1}{4}}} \tag{16}$$

where $W_{10}^{x'}$, $W_{10}^{y'}$, $U_{ice}^{x'}$, and $U_{ice}^{y'}$ are anomalies defined by $(W_{10}^x - \acute{W}_{10}^x)$, $(W_{10}^y - \acute{W}_{10}^y)$, $(U_{10}^x - \acute{U}_{10}^x)$, and $(U_{ice}^y - \acute{U}_{ice}^y)$, and $\acute{U}_{ice}^x$, $\acute{U}_{ice}^y$, $\acute{W}_{10}^x$, and $\acute{W}_{10}^y$, are sea ice velocities of zonal and meridional direction and mean values of 10 m wind, respectively. Finally, the $C_{D,io}$ and $C_{D,ai}$ can be calculated from $N_a$, $R_o$ and the mean sea ice thickness[58,59] in spring and autumn from 2004 to 2007 as obtained from NASA's Ice, Cloud, and land Elevation Satellite (ICESat) laser altimetry (Supplementary Fig. 4):

$$C_{D,io} = \frac{\rho_{ice} H f}{\rho_w R_o N_a |W_{10}|} \text{ and } C_{D,ai} = \frac{\rho_w C_{D,io} N_a^2}{\rho_a} \tag{17}$$

**Ocean surface stress curl at the air–ocean and ice–ocean**. In the coastal region, the horizontal imbalance of energy from the atmosphere into the ocean can have a large impact on ocean circulation. Therefore, ocean circulation will be strengthened by horizontal variations in the wind field and sea ice condition at the boundary of the polynya. Ocean surface stress is calculated as a combination of sea ice and wind stress as

$$\tau_o = A\tau_{io} + (1 - A)\tau_{ao} \tag{18}$$

where $A$ is the portion of the area occupied by sea ice and $\tau_{ao}$, $\tau_{io}$ are the ocean surface stress at the air–ocean and ice–ocean interface, respectively[60,61]. $\tau_{ao}$ was calculated by

$$\tau_{ao} = (\tau_{ao}^x, \tau_{ao}^y) = \rho_a C_{D,ao} |W_{10}| W_{10} \tag{19}$$

where subscript $x$ and $y$ are the zonal and meridional components, respectively; $\rho_a$ and $W_{10}$ are the air density (1.29 kg m$^{-3}$) and wind velocity vector at 10 m above the sea level, respectively. The air–ocean drag coefficient ($C_{D,ao}$) was applied depending on wind speed as[62]

$$C_{D,ao} = \begin{cases} 1.2 \times 10^{-3} & W_{10} < 11 \text{ m s}^{-1} \\ (0.49 + 0.065 W_{10}) \times 10^{-3} & 11 \leq W_{10} \leq 25 \text{ m s}^{-1} \end{cases} \tag{20}$$

ice–ocean surface stress ($\tau_{io}$) was calculated as follows:

$$\tau_{io} = (\tau_{io}^x, \tau_{io}^y) = \rho_w C_{D,io} |U_{ice} - U_w| (U_{ice} - U_w) \tag{21}$$

where $\rho_w$ is the surface water density (1026 kg m$^{-3}$); $U_{ice}$ is the sea ice velocity (observation)[53]; and $U_w$ is the water velocity[61,63]. Understandably, the water velocity was unknown below ice coverage in the AS. Thus, we estimated a parameterization assuming that a full Ekman spiral developed below the ice and that no other forces than the stress were acting on the water; this gave the surface current

velocities as[61,64,65]

$$\begin{pmatrix} U_W^x \\ U_W^y \end{pmatrix} = \begin{pmatrix} \cos\left(\frac{\pi}{4}\right) & -\sin\left(\frac{\pi}{4}\right) \\ \sin\left(\frac{\pi}{4}\right) & \cos\left(\frac{\pi}{4}\right) \end{pmatrix} \begin{pmatrix} \frac{\tau_o^x}{\sqrt{\rho_w^2 f A_Z}} \\ \frac{\tau_o^y}{\sqrt{\rho_w^2 f A_Z}} \end{pmatrix} \tag{22}$$

where $f$ is the Coriolis parameter, $A_Z$ is the vertical eddy viscosity of 0.1 m$^2$ s$^{-1}$, and $\tau_o^x$, $\tau_o^y$ are the surface stresses. The surface current velocity was calculated by repeating the above equations, starting at the surface with a current velocity of zero and continuing until convergence. Finally, ocean surface stress curl ($\tau_c$) was written as

$$\tau_c = -\left(\frac{\partial \tau_o^x}{\partial y} - \frac{\partial \tau_o^y}{\partial x}\right) \tag{23}$$

## Data availability

The mooring data that support the findings of this study are available from the corresponding author upon reasonable request. The mooring data are available from Korea Polar Data Center (KPDC) website (https://kpdc.kopri.re.kr/search/06a59de7-c931-4d0e-8622-e6c1f7d61dfd) and can request data sharing to the administrator. The CTD data (seawater temperature, salinity and dissolved oxygen) are available from KPDC website (ANA04B: https:https://kpdc.kopri.re.kr/search/9ac8e1ac-a263-40dc-be6b-529f45855c53; ANA06B: https://kpdc.kopri.re.kr/search/0edae2a7-d680-47bc-84f6-233d26c1d3a6). The model data for wind are available from the AMPS website of Ohio State University (http://polarmet.osu.edu/AMPS/) and the NCAR website (https://www.earthsystemgrid.org/dataset/ucar.mmm.amps.html). The sea ice concentration data from AMSR-E, SSMIS and AMSR-2 are available from the Sea Ice Remote Sensing website of the University of Bremen (https://seaice.uni-bremen.de/data/). Polar Pathfinder Daily 25 km EASE-Grid Sea Ice Motion Vector, Version 4 data are available from NSIDC (National Snow and Ice Data Center) website (https://nsidc.org/data/NSIDC-0116/versions/4). The sea ice thickness data measured from ICESat are available from the NASA website (https://earth.gsfc.nasa.gov/cryo/data/antarctic-sea-ice-thickness). The reanalysis model data for net surface heat flux and freshwater flux are available from the Southern Ocean State Estimate (SOSE) website (http://sose.ucsd.edu).

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

## Acknowledgements

The authors thank the officers, crew and scientists of the R/V Araon. Hee Won Yang and Tae-Wan Kim were supported by the Korea Polar Research Institute (KOPRI) (PE21110).

## Author contributions

H.W.Y. led the analysis and wrote the manuscript. T.-W.K. designed the research and contributed to the data analysis. P.D., A.K.W., A.J., H.K.H., C.S.K., K.-H.C., T.P., S.H.L., and Y.K.C. were responsible for data collection and initial data processing. All authors contributed to writing the manuscript.

## Competing interests

The authors declare no competing interests.
