## [Peer Review File · Nature Communications]

Seasonal variability of ocean circulation near the Dotson Ice Shelf, AntarcticaReviewers' Comments:

Reviewer #1:

Remarks to the Author:

"Seasonal variability of ocean circulation near the Dotson Ice Shelf, Antarctica" for Nature Communications.

This manuscript describes observation-based calculations and conclusions on heat fluxes beneath an Antarctic ice shelf. It uses substantial hydrographic data in conjunction with some bulk calculations with input from weather models and remote sensing to arrive at a picture of the variability in heat transport next to a modest-sized ice shelf cavity.

First off it must be acknowledged that this is an important and very hard-won dataset in a key location for present day thinking around the interaction of oceans and ice and implications for sea level rise. It is impressive to recover multi-year data from multiple locations as this really is a remote location. This leads to an issue that at no point do I get a sense if the system is (i) simply interesting, (ii) a key component of the important Amundsen sector or (iii) vitally important on its own. Obviously (i) is true but it would be good to hear a more active voice about why these data are important. For example L41 the melt rate of 2.6 m/yr is not so large?

Following from this, a little more specificity around motivating questions (L46-49) would help. It would be good for example to understand why temporal variability is important seeing as we are looking at long term climate impacts. Being provocative, does it matter if it seasonally varies? Or does missing out on this variability mean we get the melt rate wrong?

The bulk of the material is lumped into a large somewhat meandering Results section. Looking at Nature Comms format it looks to me like there is room for an additional section and heading and the heading text can be less generic.

The closing statement Line 169-171 suggesting that seasonal varying heat transport is previously unknown will need some clarification as Stewart et al 2019 and Malyarenko et al 2019 for example looked at a version of this topic but in the far larger Ross cavity.

Which raises an important point – how general are these results like to be? Will similar things be happening in other Amundsen Sea shelf systems? Will they be happening in other small shelf systems elsewhere? Can we expect to see anything like this in the large cold cavity systems of the Filchner-Ronne or Ross?

Their final concluding point (L170) notes that progress needs data from inside cavity systems. I agree with this strongly.

However, in a small system like this there must be a chance with such good external data that a model should do quite well. What are the options for taking the data here (hydrography, wind and sea ice) and providing a numerical solution? I imagine someone in the wider team is working on this.

I do wonder what is being lost with all these datasets missing out on the near surface data. I understand 100% why and we all face the same challenge but it calls into question exactly how robust conclusions can be. Is it worth a targeted section in the methods comparing upper SBE37s and satellite derived SST?

I struggled greatly with Fig 1. I get it that these sorts of papers have these sorts of figures but this one is particularly tough. There is all this information, but I don't have a good sense of what the cavity is like. I don't know what the 3D progressive vector diagram tells us – they can be misleading and ambiguous. I prefer 2D figures for this, colour coded with depth and tick marks every 60 days or

something similar?

Then the vitally important TS diagram in Fig 1 d/e is a blurry spec. I suggest salinity be labelled and described as Practical Salinity S_p as there is growing usage of TEOS-10 absolute salinity. We are all struggling with keeping up to date for comparable to history data so I don't have a strong preference (well ok may S_A would be better) but it should be clearly labelled.

It seems to me the really important figure is an extension of Fig 2C. Can this be separated out expanded so it is actually legible and then arrows for seasonally specific heat fluxes added in? This would really capture the message of the manuscript nicely.

Unfortunately, I think clarity around Figs 1e/d and Figs 2C come at the cost of moving Fig 2 a/b and Fig 3 to the supplementary section.

Minor

Line 121 – “influence the general oceanic circulation” ... do they mean local circulation?

Methods line 197 – this is to replace missing data?

References

Malyarenko, A., Robinson, N.J., Williams, M.J.M. and Langhorne, P.J., 2019. A wedge mechanism for summer surface water inflow into the Ross Ice Shelf cavity. *Journal of Geophysical Research: Oceans*, 124(2), pp.1196-1214.

Stewart, C.L., Christoffersen, P., Nicholls, K.W., Williams, M.J. and Dowdeswell, J.A., 2019. Basal melting of Ross Ice Shelf from solar heat absorption in an ice-front polynya. *Nature Geoscience*, 12(6), pp.435-440.

Reviewer #2:

Remarks to the Author:

I found the manuscript by Yang et al. very interesting both in terms of the quality of the observations (winter data from the Amundsen Sea are particularly exciting given the difficulty to obtain them) and proposed mechanism to explain seasonality in ocean heat transport toward the Dotson Ice Shelf. The impact of wind curl near the coast on ocean heat delivery to Antarctic glaciers is poorly understood, and this study shows that it can be important. However, the study does not describe important processes that happen in the Amundsen Sea that could cause/contribute to the observed seasonality, including the role of coastal polynyas. Therefore, I believe that these other aspects need to be analysed before we can safely conclude that ocean surface curl is the (main) driver of the observed seasonality. Below I have included some major and minor comments.

Major comments

- In front of the Dotson Ice Shelf there is a polynya. Convection driven by enhanced winter sea ice formation in polynyas plays an important role in setting the properties of the water that can reach the ice cavity. A discussion/quantification of the role of this polynya is missing here and needs to be included in order to confidently conclude that ocean surface curl is the main driver of the seasonality.
- Comparison with previous studies (e.g. Webber et al., 2017; *Nat Comm*) showing seasonality near other glaciers in the Amundsen Sea (e.g. Pine Island Glacier) is missing. I strongly encourage to compare your results with previous work.
- The seasonality includes two peaks in ocean heat transport toward the Dotson Ice Shelf (see Fig. 4).

One peak in summer and one in winter. The winter peak is not discussed in the text. I do recommend further analysis to understand the driver of this second peak, which might be related to winter warming in the bottom layer.

Minor Comments

- Line 44: use either mCDW or CDW in the text.
- Figure 1b/c. Increase font size for visibility.
- Line 69: "cold and fresh water".
- Figure 2: "Uphill side of white hatched line". Please rephrase the sentence.
- Figure S2: in the caption -> "Time series of temperature and salinity at....".
- Line 81-84: The seasonality in salinity at mid-depth seems related to thermocline depth variability. Please clarify this.
- Figure 3e: Please specify in the caption the lag between the two lines.
- Line 112: It seems that correlation between OSSC and meridional ocean velocity in the western flank has no lag, while there was a lag of 35 days in the eastern flank. Is there a reason for this difference? Also can you say few more word about how a barotropic process (OSSC forcing) requires 35 days to see a response in the currents. I would expect a relatively quick response.
- Line 120: I would clarify here what meltwater fraction is (e.g. concentration of glacial meltwater in seawater).
- Line 121: what do you mean by "long time scales"? years? Please clarify.
- Line 135: "At deeper depth"->"deeper in the water column".
- Line 136:"Countervailed"->"countervailing"?
- Line 129-140: There are many processes described here. I would explain them more in detail, maybe in two separated paragraphs.
- Line 148: remove "that".
- Line 143: Looking at Fig.4 it seems that there are two peaks in heat transport, one in summer and one in winter. I agree that the summer peak is stronger, but it could be argued that the seasonality is more complex than what is highlighted in the main text. Indeed from Fig. 4a you can see warming during winter.
- Fig S3: be clear in the caption that here you are showing depth changes of temperature/salinity "contours".
- Line 159: Conclusions/Discussion/Summary heading is missing.

Reviewer #3:

Remarks to the Author:

Review of NCOMMS-21-02666

The manuscript titled "Seasonal variability of ocean circulation near the Dotson Ice Shelf, Antarctica" by Yang et al. describes new results from multi-year moorings in front of one of the most important ice shelves from a susceptibility and potential mass balance perspective. Even their basic observations of heat content and subsurface current speeds would be noteworthy, but the tandem use of moorings on both sides of the main trough allow for an even deeper look into inflow-outflow links, which provides some of the best estimates of heat fluxes through a sub-ice shelf cavity to date. Their additional and novel link with surface curl, particularly accounting not just for wind driven curl but also that done by sea ice, provides additional insight into the forcing mechanism. Their methodology appears to be sound and well presented, more thorough than a typical manuscript of this type. The work seems to be reproducible, provided that all the necessary data can be obtained in a similar format. This leads me to one Major Comment – I see no links to where the mooring data will be

available. If not the raw files, the standard within the community these days is at a minimum that data needed to reproduce the figures in the main text. Did I miss this? Are the authors waiting until acceptance?

Overall, I think this manuscript is close to publication, after addressing my broad comments above and this (admittedly lengthy) list of minor comments.

Minor Comments:

Line 37-39: "deep-draft ice is more readily accessible" – *more* readily accessible compared to where? Rest of Antarctica? If so, why? Shelf bathymetry? Atmospheric forcing?

Line 45: "leading to mass loss" – but there already is mass loss of DIS, so your use of 'stronger' implies an 'increase' in mass loss.

Line 46: "of that circulation" – What circulation is "that" referring to? Across-, or mid-shelf?

Line 53: "is" to "are"

Line 63-64: "... the current was somewhat weaker" – is this expected?

Line 67: "the freezing point" – although stated in the methods, maybe mention here that T_f is function of both S and P.

Line 69: "relatively cold and freshwater" – change to 'fresh water'. Also how much colder and fresher? 'Relative' is too vague.

Line 71-72: "depth-independent" – barotropic?

Line 74: did you quantify and compare their periodicity, or is this hand-waving?

Line 77: remove "the"

Line 85: "the salinity difference ..." – what are implications of this? Level of non-divergence? Otherwise, if not important/relevant, consider dropping.

Line 91-93: "To estimate ..." - The start of this paragraph comes out of nowhere. E.g. you haven't said why OSSC would relate to barotropic variability. Seems like there needs to be an overview/motivation precursor to this "such and such was calculation" sentence.

Line 106: "southeastward" – consistent hyphenation of these cardinal directions. Sometimes used, sometimes not.

Line 120: Missing space after period?

Line 125: How does 1% compare to estimates from other ice shelves in Amundsen Sea embayment or elsewhere? It doesn't sound like much, so some context will help the reader.

Line 129: "inflow" I expect you mean this to be an inflow into the water column at the mooring, but can also be read as an inflow into the sub ice-shelf cavity.

Line 130: "can"? - Do you mean "did"?

Line 131: "plotted" – In a "letters"-style journal with condensed and limited text, avoid sentences describing what was plotted. A more interesting and compelling sentence will describe the process of observation, and the reader can follow the figure citation to follow-along and decide for themselves.

Line 136: This sentence is awkward, including inconsistent tense. Either ", countervailing" or maybe "... that countervailed".

Line 137: "Furthermore, ..." – very general, not saying very much. Made worse by use of "could" and ending with "again". I'd drop it or make it more specific and precise to your findings.

Line 138-140: "Finally, the decrease of meltwater discharge in spring may..." – Reader would benefit from sentence on the basic result here (along with reference to the figure), before jumping into its effect.

"Homogenize" needs an "s"?

Can also change "may" to "would", and then adding a citation would be appropriate, perhaps this one?

Silvano, A., Rintoul, S. R., Pena-Molino, B., Hobbs, W. R., van Wijk, E., Aoki, S., Tamura, T., & Williams, G. D. (2018). Freshening by glacial meltwater enhances melting of ice shelves and reduces formation of Antarctic Bottom Water. *Science Advances*, 4, eaap9467.
<https://doi.org/10.1126/sciadv.aap9467>

Line 146: "Not observed in remote sensing..." and "unaccounted for" – really? I know several individuals who would take offense from this sentence. Consider stereophotogrammetry-based DEM differencing from PGC, recent ICESat-II results, moorings in front of ice shelves, etc., etc. Either be more specific to make it correct, or reign in this (specific) claim of novelty.

Line 148: remove "that"?

Line 150-153: "Nevertheless, the ..." – surely it will propagate, but what is the relative size of this effect? Hard to say I'm sure, but the reader may wonder how important this effect will be.

Line 152: drop "s" from "propagates"?

Line 153: drop "s" from "DIS's"?

Line 154-158: Is there a physical reason why a 74 days lag would result in a peak correlation? Does any dynamic theory, or modeling work? Also, I recall some drifting floats advecting under and out of some ice shelves in the region. What was their 'residence' time?

Girton, J. B., Christianson, K., Dunlap, J., Dutrieux, P., Gobat, J., Lee, C., & Rainville, L. (2019). Buoyancy-adjusting Profiling Floats for Exploration of Heat Transport, Melt Rates, and Mixing in the Ocean Cavities Under Floating Ice Shelves. *OCEANS 2019 MTS/IEEE SEATTLE*, 1–6.
<https://doi.org/10.23919/OCEANS40490.2019.8962744>

Also, I'm curious what the correlation looked like for the whole range of lags. Is it a sharp max at 74 days, or more broad peak? This would tell the reader how significant this "74" is.

Line 163: "relates" – you've spent a lot of time arguing causation, so be specific here, or in danger of confusing reader or obfuscating this key (and well supported) finding

Line 167: "Understanding these processes..." – a throwaway sentence. This can be said about nearly any Earth process. Be more specific (e.g. what exactly needs to be better understood to understand

the long-term trend, and for what reason) or dedicate this precious space to something else.

Line 170: "... previously unknown ..." – similar to my comments to Line 146, this is not an unknown effect (I presume you are referring to the seasonally varying heat transport), thinking of recent work in Ross, Filchner-Ronne, and Amundsen. I am most excited by the west-vs-east differences, the influx-outflow lag, baroclinic-vs-barotropic, and OSSC findings – which would be better to highlight in these closing overview sentences.

Methods section: There are some subsection headers, but I still had a hard time keeping track with the purpose of each little bit of text and equations. Perhaps sub-subsections would help.

Line 197: regarding method of obtaining velocities below 600m at K4 – will seasonality bias your method at all?

Also, Lines 200-203 – these regression equations didn't do much for me. Are they needed? I think it's enough to show Supp Figure 4.

Line 215: "especially" – don't know why this word is used.

Line 223: "low layer" – ambiguous - what depths does this refer to?

Line 257: "if U_{ice} assumed to be far larger than U_w " – is this a good assumption? Not good enough for me to not include a citation or other evidence.

Line 276: h_{ice} is taken from ICESat – does the interannual variability of h_{ice} preclude you from using this method?

Figure 1 and throughout: do you need to label "salinity" with units?

Figure 1a – is the contouring of overview map ocean depths, or ocean temperatures (it appears unlabeled)? Seems a difference color map than other contours.

Figure 2: although K4 is mentioned, remind readers whether it is on East or West flank.

Supp Figure 3 caption: Leave out the "show a good correlation", if indeed a correlation was actually done – either way, this is a result and should be in the text. The figure captions are for describing what is plotted only.

Supp Figure 4b: The HF variability in this time series is so high that I'm not sure what is supposed to be communicated to the reader. LP filter? Zoom in on some period?

Supp Figure 5: c and d panels don't tell me much. I'd recommend dropping if not in Supplement.

Response to reviews of “Seasonal variability of ocean circulation near the Dotson Ice Shelf, Antarctica”

Manuscript NCOMMS-21-02666

Response to Reviewers

We would like to thank the reviewers and editors for their time and effort in reviewing the paper "Seasonal Variability in Ocean Circulation Near the Dawson Ice Shelf, Antarctica," submitted for publication in Nature Communications. Efforts were made to accommodate reviewers' comments in the resubmitted revision, and most comments reflect. For some of the comments that do not reflect, we provided justifications and detailed explanations for them. Changes are highlighted in the manuscript. See step-by-step answers to reviewers' comments below. Here, all line numbers refer to revised manuscript files.

Reviewer #1.		
ID	Comment Major	Response
1	It is impressive to recover multi-year data from multiple locations as this really is a remote location. This leads to an issue that at no point do I get a sense if the system is (i) simply interesting, (ii) a key component of the important Amundsen sector or (iii) vitally important on its own. Obviously (i) is true but it would be good to hear a more active voice about why these data are important. For example L41 the melt rate of 2.6 m/yr is not so large?	We agree with Reviewer's opinion. It seems that the point of this study was lacking. We cited existing studies and added sentences to the introduction to explain why the Amundsen Sea is essential and why this study is necessary. Please see the added and revised sentences listed below.  1. Edit Line 37-39: “In the Amundsen Sea (AS), where the seabed depth of the continental shelf is sufficiently deep (>500 m), Circumpolar Deep Water (CDW) intrusion over the slope under the influence of Ekman transport and flow to the south along the deep trough.” 2. Edit Line 41-51: “The oceanic heat transport in the AS is more significant than the Ross and Weddell Seas adjacent to the giant ice shelf in Antarctica. In the Ross and Weddell Seas, where large cyclonic polar gyres are located north of the continental shelf, only a tiny amount of CDW cooled within the gyres intrusion the continental shelf. Understanding seawater circulation near the ice shelf is essential for determining how changes in oceanic heat transport affect basal melting of ice shelves. Recently, the effect of surface water flowing into the cavity on the melting of the Ross and Filcher-Ronne ice shelves and its seasonality had been studied. Some studies in front of Pine Island Glacier (PIG) have shown that the long-term variation in sea surface heat flux can affect the variability in CDW volume and seawater circulation pattern. Moreover, the seasonal expansion and contraction of Polynya and the sea ice melting and formation can affect the seawater density structure, affecting the seawater circulation.” 3. Edit Line 53-63: “Such rate fast compared to the average thickness change in the AS sector (1.94 m yr^{-1}) and 12 times faster than the Ross Sea on average (0.21 m yr^{-1}). Previously research has indicated a seasonal varying of mCDW intrusions near the entrance of the trough connected to DIS due to changes in sea ice and wind distribution. In front of DIS, the prior observed data are insufficient to investigate the seasonality of mCDW circulation because most of the research has mainly

		conducted in the summer. However, it is necessary to confirm the seasonality of the mCDW circulation in front of the ice shelf to quantify the influx of oceanic heat on ice shelf melting. In addition, the identification of seasonal variation causes will contribute to understanding the process of ice shelf melting due to long-term ocean variability. In this study, we reveal the unknown seasonal variability of mCDW circulation in front of the DIS and how it propagates into the cavity beneath the ice shelf.”
2	Following from this, a little more specificity around motivating questions (L46-49) would help. It would be good for example to understand why temporal variability is important seeing as we are looking at long term climate impacts. Being provocative, does it matter if it seasonally varies? Or does missing our on this variability mean we get the melt rate wrong?	Analyzing the seasonal variation of the observations is very important in quantifying the mCDW entering the ice shelf. Some previous studies at the entrance to the trough show seasonal variability of the CDW intrusion along the DGT. However, previous studies in front of DIS were mainly conducted in summer. Therefore, the results of previous studies alone cannot quantify the oceanic heat transport into the ice shelf cavity. In addition, these results will provide a clue in understanding how long-term ocean variability propagates in ice shelf melting. To highlight the importance of our research results, some previous studies have been newly cited, and the introduction has been revised and supplemented as follows. Edit Line 55-63: “Previously research has indicated a seasonal varying of mCDW intrusions near the entrance of the trough connected to DIS due to changes in sea ice and wind distribution. In front of DIS, the prior observed data are insufficient to investigate the seasonality of mCDW circulation because most of the research has mainly conducted in the summer. However, it is necessary to confirm the seasonality of the mCDW circulation in front of the ice shelf to quantify the influx of oceanic heat on ice shelf melting. In addition, the identification of seasonal variation causes will contribute to understanding the process of ice shelf melting due to long-term ocean variability. In this study, we reveal the unknown seasonal variability of mCDW circulation in front of the DIS and how it propagates into the cavity beneath the ice shelf.”
3	The bulk of the material is lumped into a large somewhat meandering Results section. Looking at Nature Comms format it looks to me like there is room for an additional section and heading and the heading text can be less generic.	We agree with Reviewer's opinion. The parts are separated and subtitles are added. Introduction Results - Observation - Seasonal variation of mCDW near the eastern and western flank - mCDW circulation and ocean surface stress curl - Meltwater outflow - Relation between heat transport and meltwater flux Discussion
4	The closing statement Line 169-171 suggesting that seasonal varying heat transport is previously unknown will need some clarification as Stewart et al 2019 and Malyarenko et al 2019 for example looked at a version of this topic but in the far larger Ross cavity.	The purpose of this sentence is that "the mechanism of seasonal variability in the ocean circulation across the ice shelf is not well known.". In addition to the two papers (Stewart et al. 2019 and Malyarenko et al. 2019) mentioned by the reviewer, there are several previous studies (Webber et al. 2017 and Assmann et al. 2019) conducted in the Amundsen Sea about the variability of the ocean circulation in front of the ice shelf. Therefore, we have revised this sentence more clearly because this sentence may cause misunderstanding to some readers, as the reviewer mentioned, and previous studies on seasonal changes of ocean refining are cited in the text as follows.

		Edit Line 46-50: “Recently, the effect of surface water flowing into the cavity on the melting of the Ross and Filcher-Ronne ice shelves and its seasonality had been studied. Some studies in front of Pine Island Glacier (PIG) have shown that the long-term variation in sea surface heat flux can affect the variability in CDW volume and seawater circulation pattern.” Edit Line 228-231: “The circulation pattern of mCDW in the front of the ice shelf and its seasonal variability by atmospheric conditions evidenced here are potentially significant for understanding ice shelf retreat due to climate change and needs further study, e.g., ice-ocean coupled numerical model study considering the ice shelf.”
5	Which raises an important point – how general are these results like to be? Will similar things be happening in other Amundsen Sea shelf systems? Will they be happening in other small shelf systems elsewhere? Can we expect to see anything like this in the large cold cavity systems of the Filchner-Ronne or Ross?	Understanding the seasonal variability of ocean circulation is very important to quantitatively understand the impact of oceanic heat transport on ice shelf melting. In addition, understanding the effects of atmospheric variability on ocean circulation and its mechanism is very important in understanding the process of ice shelf retreat caused by long-term ocean variability in relation to climate change. We have revised the introduction as follows to highlight these areas. Edit Line 58-64: “However, it is necessary to confirm the seasonality of the mCDW circulation in front of the ice shelf to quantify the influx of oceanic heat on ice shelf melting. In addition, the identification of seasonal variation causes will contribute to understanding the process of ice shelf melting due to long-term ocean variability. In this study, we reveal the unknown seasonal variability of mCDW circulation in front of the DIS and how it propagates into the cavity beneath the ice shelf. In addition, temporal and spatial variations in ocean circulation and heat transport to and from the sub-DIS cavity and look into associated forcing mechanisms are investigated.” In this study, the fluctuation of seawater circulation due to ocean surface stress curl due to wind and sea ice motion explains. In general oceans, the occurrence of ocean circulation due to convergence and divergence of surface seawater causes by a spatial imbalance of wind stress curl are very common phenomena. In particular, strong winds parallel to the coastline on the coasts bordering the continent create the coastal upwelling and seawater circulation. Previous studies on the Antarctic coast have done a lot of research on the relationship between wind-induced Ekman pumping and CDW thickness variability. In particular, such studies actively conduct in the Amundsen Sea, where a significant amount of CDW was introduced into the continental shelf (Dutrieux et al. 2014; Wåhlin et al. 2013; Kim et al. 2017; Webber et al. 2017; Davis et al. 2018; Assmann et al. 2019). However, most of the previous studies focused on the change in CDW thickness by Ekman pumping. They did not get enough evidence that this could affect the variability of the seawater circulation. In front of Pine Island Glacier (PIG), studies have conducted that the annual variability of buoyancy flux at the ocean surface can affect ocean circulation (Webber et al., 2017). We believe that it is necessary to confirm whether the seawater circulation due to atmospheric variability applies to other small-shelf systems. And long-term high-resolution data need for this. Implementing a numerical model can be a good alternative. In

		addition, there is a sufficient possibility of such a seawater circulation occurring in the front of big ice shelves such as the Filchner-Ronne or Ross ice shelf. However, compared to the Amundsen Sea, the amount of CDW inflow is small, and it is thought that oceanic heat transport to the ice shelf due to seawater circulation will not be sufficient.
6	However, in a small system like this there must be a chance with such good external data that a model should do quite well. What are the options for taking the data here (hydrography, wind and sea ice) and providing a numerical solution? I imagine someone in the wider team is working on this.	Previous model studies (Yoshihiro Nakayama and Satoshi Kimura) have been conducted focusing on the basin scale circulation of the Amundsen embayment and was especially utilized to confirm the importance of the role of ocean circulation in the continental shelf onshore transport and offshore. Therefore, as in this study, the scale of the model performed in the past is too large to investigate the ocean circulation by local processes in front of the ice shelf. In the near future, we intend to proceed with a study to identify this process using a fine resolution model that more precisely considers the cavity structure and the topography in front of the ice shelf.
7	I do wonder what is being lost with all these datasets missing out on the near surface data. I understand 100% why and we all face the same challenge but it calls into question exactly how robust conclusions can be. Is it worth a targeted section in the methods comparing upper SBE37s and satellite derived SST?	As you said, the measurement of the sea surface data from ocean mooring in the adjacent sea of the ice shelf is a risky challenge. For example, in front of the Dotson Ice Shelf is a relatively low occurrence frequency of giant icebergs compared to Pine Island Bay and Thwaites Glacier, but icebergs with a height of several hundred meters often appear. Therefore, to reduce damage to mooring equipment from icebergs, the top float of mooring was designed to be located 250m below in the water. Unfortunately, as a result, we can't obtain data from the surface layer to 250m. The CDW inflows along the eastern slope were located under the 400m depth during the entire observation period, and it could be entirely observed from K4 mooring. However, as shown in the figure below (Shipboard CTD), the meltwater discharge through the western slope is mainly located on the upper 400m. Therefore, the quantitative estimate of the meltwater discharge from the K5 mooring installed on the western slope is insufficient.

Fig. Calculated meltwater fraction from the shipboard CTD data (temperature, salinity, dissolved oxygen) measure in January 2014 and 2016 at the western mooring station(K5). The cyan box indicates the covered water column by K5 mooring.

As you suggest, we compared the seawater temperature obtained from SBE37 (270m) with the SST obtained from OISST (Optimum Interpolation Sea Surface Temperature obtained from satellites, ships, buoys, and Argo floats) near the Mooring data (K5). The SST showed seasonal variation due to heat exchange with the atmosphere, but it did not correspond well with observed seawater temperature at 270m depth.

Fig. Comparison between SST obtained from the OISST and seawater temperature observed using SBE37 at 270m of mooring K5.

8

I get it that these sorts of papers have these sorts of figures but this one is particularly tough. There is all this information, but I don't have a good sense of what the cavity is like. I don't know what the 3D progressive vector diagram tells us – they can be misleading and ambiguous. I prefer 2D figures for this, colour coded with depth and tick marks every 60 days or something

We modified it to increase the visibility of Figure 1. Figures 1d and e enlarged, and the complex 3D progressive vector diagrams of Figures 1b and c have been changed to 2D and marked the black cross on the diagram every 3months interval.

	similar?	
9	Then the vitally important TS diagram in Fig 1 d/e is a blurry spec. I suggest salinity be labelled and described as Practical Salinity S_p as there is growing usage of TEOS-10 absolute salinity. We are all struggling with keeping up to date for comparable to history data so I don't have a strong preference (well ok may S_A would be better) but it should be clearly labelled.	You fully agree with the comments. We have specified the unit of salinity (PSU) in all parts of the paper as well as in Figures 1 d and e.
10	It seems to me the really important figure is an extension of Fig 2C. Can this be separated out expanded so it is actually legible and then arrows for seasonally specific heat fluxes added in? This would really capture the message of the manuscript nicely.	As shown in Figure 2b, the southward velocity increases toward the bottom, but the variability does not change significantly with the depth. The depth-averaged southward flow shows both seasonal and intra-seasonal variability. The southward flow corresponds well with the variability at the OSSC calculation point (74°S, 112.25°W) closest to the mooring station K4. In other words, it can say that the OSSC is the dominant force for the variability of mCDW intrusion to the ice shelf cavity. Figure 2c shows the spatial distribution of seasonal averaged ocean surface stress (blue arrow) and ocean surface stress curl over eight seasons during two years, together with the sea ice distribution. Also, we newly added the seasonal average of the southward velocity and heat transport calculated from Eq. (2) in the figure. Moreover, we add the new sentence as follows to explain these results. Edit Line 139-142: "In addition, the increase of OSSC on the eastern flank in summer leads to enhance the heat transport to the ice shelf by causing the strengthen of southward mCDW flow. Conversely, both southward flow and heat transport decrease in winter."
11	Unfortunately, I think clarity around Figs 1e/d and Figs 2C come at the cost of moving Fig 2 a/b and Fig 3 to the supplementary section.	We modified the T-S diagram in Figure 1 and the horizontal distribution of OSSC in Figure 2 for good visibility without separating them. Figure 2a, b shows the temporal variation of southward mCDW flow, and Figure 3 explains the relationship between the seawater flow and the meltwater fraction along the western slope, and since it is often cited in the text, this is kept as it is.
12	Line 121 – "influence the general oceanic circulation" ... do they mean local circulation?	The previous ambiguous sentence has changed to the following. Edit Line 165-166: "The meltwater outflows occurred mainly in the middle layer and may be significant enough to influence the density structure in the water column and further regional seawater circulation."
13	Methods line 197 – this is to replace missing data?	Yes. Unfortunately, one 300 kHz ADCP for observing the low layer current velocity at the K4 station has stopped working since January 2015. However, we wanted to calculate the amount of heat transport to the ice shelf over the entire period. For this, as shown in Supplementary Figure 8, we compared the one-year current velocity data recorded by stopped ADCP with current velocity data at 600 m depth observed by another 150 kHz ADCP of the K4 mooring station. Furthermore, we got the high correlation and estimated the missed current velocity data from these correlations. Finally, we calculated the amount of heat transport during the entire observation period by estimating

		the current velocity in the bottom layer during the missed one year.
--	--	--

Reviewer #2.		
ID	Comment Major	Response
1	In front of the Dotson Ice Shelf there is a polynya. Convection driven by enhanced winter sea ice formation in polynyas plays an important role in setting the properties of the water that can reach the ice cavity. A discussion/quantification of the role of this polynya is missing here and needs to be included in order to confidently conclude that ocean surface curl is the main driver of the seasonality.	We agree with your opinion. The extensive Polynya develops in front of DIS in summer, and Polynya exists only in a minimal area in winter. Repeated seasonal contraction and expansion of Polynya can affect seawater properties and circulation. First, since the stress intrusion into the ocean from the atmosphere is blocked by sea ice and its effect change, the spatial distribution of sea ice causes a spatial imbalance in the stress of the ocean surface under the homogenous wind field. Therefore, the OSSC at the boundary of Polynya shows a relatively large amplitude. Second, the melting and formation of sea ice during the expansion and contraction of polynyas causes salinity changes in the ocean surface. When sea ice melts, a large amount of fresh water supply into the ocean to strengthen stratification, and when sea ice is formed, the salinity of the surface layer increases due to brine rejection, deep convection is strengthened. We have further explained the effect that the seasonal variability of Polynya can have on the ocean in front of DIS. The buoyancy flux calculates and discusses the impact of sea ice formation and melting in the text as follows and adds the calculation result of buoyancy flux in supplementary fig. 3a. Edit Line 50-51: “Moreover, the seasonal expansion and contraction of Polynya and the sea ice melting and formation can affect the seawater density structure, affecting the seawater circulation.” Edit Line 113-123: “In the north of DIS, Amundsen Sea Polynya (ASP) repeats seasonal expansion and contraction, which can be affected seawater circulation and spatial distribution of mCDW by causing changes in ocean surface density during sea ice formation and melting. The previous study suggests that positive buoyancy flux at the sea surface by sea ice formation and surface cooling creates deep convection, and the mCDW layer descends in front of PIG. In contrast, negative buoyancy flux due to surface heating and sea ice melting results in increase mCDW volume. To investigate the effect of sea ice fluctuations on the change of mCDW and its circulation, buoyancy flux was calculated using the heat and freshwater flux at the sea surface and compared with fluctuation in the depth of isohaline (Supplementary Fig. 3a). Both buoyancy flux and salinity show the seasonal variation that increased in summer and decreased in winter. However, in intra-seasonal variability, they do not agree well; thus, the effect of buoyancy flux on mCDW variability and circulation is expected to be insignificant.” Edit Line 124-126: “At the boundary of ASP, the steep gradient of sea ice concentration causes the spatial stress imbalance into the ocean under the homogenous wind field. As a result, it

		causes the divergence or convergence of the sea surface, and spatial difference of sea surface elevation can be induced barotropic current.”
2	Comparison with previous studies (e.g. Webber et al., 2017; Nat Comm) showing seasonality near other glaciers in the Amundsen Sea (e.g. Pine Island Glacier) is missing. I strongly encourage to compare your results with previous work.	We have accepted your opinion. Therefore, we explain and cite the Webber et al., 2017 study in Pine Island Bay in the introduction and text as follows. Edit Line 48-50: “Some studies in front of Pine Island Glacier (PIG) have shown that the long-term variation in sea surface heat flux can affect the variability in CDW volume and seawater circulation pattern” Edit Line 115-117: “The previous study suggests that positive buoyancy flux at the sea surface by sea ice formation and surface cooling creates deep convection, and the mCDW layer descends in front of PIG.” And we review the effect of the buoyancy flux on long-term variability of the CDW layer and the ocean circulation suggested by him. For this, we calculate the buoyancy flux (see the method) used the surface heat and freshwater flux at the ocean surface calculated in the Southern Ocean State Estimate (SOSE) model. The calculated buoyancy flux at the point adjacent (74.125°S, 112.25°W) to the K4 mooring station compared with the southward velocity and isohalines (supplementary fig. 3a). As a result, similar to the calculation results of Sun & Stewart 2016, the buoyancy flux was mainly influenced by the freshwater flux due to the melting and formation of sea ice. It showed distinct seasonal variability with isohalines and southward velocity. However, in its intra-seasonal variation, it was different from that of current velocity and isohaline. That is, as suggested by Webber et al., 2017, the buoyancy flux may be one of the forcings that can explain the long-term variability of CDW circulation and CDW layer, but the buoyancy flux is not sufficient to explain the short temporal variability of the southward flow in front of DIS. Reference Sun, S., Eisenman, I., & Stewart, A. L. (2016). The influence of Southern Ocean surface buoyancy forcing on glacial interglacial changes in the global deep ocean stratification. Geophysical Research Letters, 43(15), 8124-8132.
3	The seasonality includes two peaks in ocean heat transport toward the Dotson Ice Shelf (see Fig. 4). One peak in summer and one in winter. The winter peak is not discussed in the text. I do recommend further analysis to understand the driver of this second peak, which might be related to winter warming in the bottom layer.	The two peaks of winter heat transport (June and July 2014, May and June 2015) you mentioned are mainly due to the increase of the southward velocity. The southward heat transport calculates from the heat content and velocity. That is, the strengthening of the southward velocity in winter increases the heat transport. Since we showed that OSSC influences the southward velocity, we performed additional analysis on OSSC variability to find the cause of the winter heat transport peak. As shown in added newly supplementary figure 6, OSSC increased in May, June 2014, April, and May 2015, and the wind strength markedly increased during this period. Therefore, it can interpret that a peak occurs in heat transport because the OSSC is increasing due to the stronger wind, and the southward velocity strengthens. We add the following sentences to the text: Edit Line 196-199: “Although the peak of heat transport in winter was smaller than that in summer, it was conspicuous (July 2014 and June 2015, Fig. 4b). Such two winter peaks

		mainly depend on the strengthening of the southward flow and are related to the increase of OSSC due to the strengthening of the wind (Supplementary Fig. 6).” Meanwhile, we investigated your proposed winter warming in the bottom layer. During the two winter peaks, the heat content increased in the entire layer, and the increase was more significant in the middle layer (400-600 m) than in the lower layer. The meridional gradient of temperature and salinity (TS) in the middle layer is greater than that of the deep layer. The effect of downwelling or upwelling in front of the ice shelf on TS changes is more significant in the mixed layer between WW and CDW, where the vertical gradient of TS is larger than in the bottom layer. Also, the strong downwelling (upwelling) in the front of the ice shelf generates a positive (negative) gradient of TS in the meridional direction (Kim et al., 2016), so the southward flow causes an increase (decrease) in the TS. In summary, the effect of the barotropic southward flow on the heat content variability greater in the middle layer than in the bottom layer. It can confirm by decreasing the fluctuation range of the isohaline and isotherm lines (supplementary figure 3) and heat content line (below figure) toward the bottom.  Fig. Timeseries variation of heat content lines
	Minor	
4	Line 44: use either mCDW or CDW in the text.	Done. In the revised manuscript, mCDW was defined separately as follows. Edit Line 40: “... by warm and salty modified CDW (slightly colder and fresher water compared with CDW; mCDW)...”.
5	Figure 1b/c. Increase font size for visibility.	I took the comments of other reviewers as well, corrected the fig 1b/c, and increased the size of the font.
6	Line 69: “cold and fresh water”.	Done. the word spacing was corrected as follows. Edit Line 87: “...the outflows of relatively cold and fresh water (-1 □, 34.2 PSU) were found above 500 m near the ...”.
7	Figure 2: “Uphill side of white hatched line”. Please rephrase the sentence.	Done. That sentence has been changed to as follows. Edit Line 522: “Outside of the white line....”.
8	Figure S2: in the caption -> “Time series of temperature and salinity at....”.	Done. ‘vertical’ was deleted. That sentence has been changed to as follows. Edit Line 15 in Supplementary information: “Time series of temperature and salinity at....”.
9	Line 81-84: The seasonality in salinity at mid-depth seems related to thermocline depth variability. Please clarify this.	We observed the temperature and salinity profile using shipboard CTD when the mooring was install and recovery at the eastern slope of DIS (see figure below). We find the two sections with a large vertical temperature gradient can be identified by mixing AASW (Antarctic Surface Water), WW,

and mCDW in 2016. Since our mooring data does not cover the near-surface variability, we only focus on the lower section. A large temperature gradient mainly appears in the 400-700 depth, and the salinity distribution in this depth is 34.25~34.4 PSU in 2014 and 34.25~34.5 PSU in 2016. Therefore, if the variability of thermocline depth is the leading cause of seasonality of salinity at mid-depth, 34.3 and 34.4 PSU isohaline should show distinct seasonal variability but, they did not (see the supplementary figure 3a). Instead, 34.2 PSU isohaline showed distinct seasonal variability compared to 34.3 and 34.4 PSU isohaline, and this variability was in good agreement with the variability of the depth-averaged southward velocity. In particular, it was found that the depth of all isohalines became shallow when the winter peak of the southward velocity appeared. Therefore, it can be said that barotropic southward flows influence the seasonal variation of salinity and intra-seasonal variation.

Fig. Observed temperature and salinity profiles using shipboard CTD in January 2014 and 2016 at K4 mooring station.

10	Figure 3e: Please specify in the caption the lag between the two lines.	Done. The following sentence was added. Edit Line 534: “Upper x-axis is back-shifted by 17-day.”
11	Line 112: It seems that correlation between OSSC and meridional ocean velocity in the western flank has no lag, while there was a lag of 35 days in the eastern flank. Is there a reason for this difference? Also can you say few more word about how a barotropic process (OSSC forcing) requires 35 days to see a response in the currents. I would expect a relatively quick response	Before explaining the lags on the eastern and western slopes, we found an error while reviewing the manuscript and corrected it in the resubmitted manuscript. We performed a running average of all data for 31 days to confirm the relationship and lag between the depth-averaged meridional current velocity and OSSC in the previous manuscript. However, this may not be an excellent way to process data, as running averages can skew the lag. We first performed cross-spectral analysis to confirm that both meridional velocity and OSSC had a periodicity in the time series data of the eastern slope. As a result, significant coherence finds in a frequency of about 80 days (Supplementary Figure 5). And the 20-day low-pass filtered southward velocity and OSSC showed the highest correlation ($r=0.47$) with the 28-day lag (Fig. 2d and e). Moreover, they showed a correlation of higher than 0.4 at the 23-36 day lag. This lag is about 1/4 of the 80-day frequency, which can be interpreted that the cumulative OSSC variability influences the southward velocity. On the other hand, no statistically significant correlation was found between OSSC and meridional velocity in the western slope. We

		added explanations for these new results to the text as follows with new figures (Fig. 2d and e and Supplementary Fig. 5). Edit Line 129-135: “The variability of OSSC on the eastern flank was relatively high compared to the western flank. As a result of cross-spectral analysis between OSSC (74°S, 112.25°W) and the vertically averaged southward velocity at the eastern flank, significant coherence appeared in the around 80-day frequency (Supplementary Fig. 5). In addition, 20-day low-pass filtered southward velocity showed the highest correlation ($r=0.47$) with OSSC in the 28-day lag (Fig 2d and e), approximately a quarter of the 80-day frequency, suggesting that the accumulated OSSC affects barotropic southward flows.”
12	Line 120: I would clarify here what meltwater fraction is (e.g. concentration of glacial meltwater in seawater).	Done. That sentence has been changed to as follows. Edit Line 164: “...maximums of the meltwater fraction (e. g. concentration of glacial meltwater in seawater)...”
13	Line 121: what do you mean by “long time scales”? years? Please clarify.	As the meaning of the sentence was too broad and general, the sentence was changed in more detail as follows. Edit Line 165-166: “The meltwater outflows occurred mainly in the middle layer and may be significant enough to influence the density structure in the water column and further regional seawater circulation.”
14	Line 135: “At deeper depth”->”deeper in the water column”.	Done. That sentence has been changed to as follows. Edit Line 180: “Deeper in the water column,.....”
15	Line 136:”Countervailed”->”countervailing”?	Done. That sentence has been changed to as follows. Edit Line 183: “....baroclinic current countervailing the northward barotropic current.”
16	Line 129-140: There are many processes described here. I would explain them more in detail, maybe in two separated paragraphs.	Done. That paragraph divides into two paragraphs, and each paragraph's content has been added and modified at the request of reviewers (see the edit line 175-190).
17	Line 148: remove “that”.	Done. ‘that’ was deleted (see the edit line 202).
18	Line 143: Looking at Fig.4 it seems that there are two peaks in heat transport, one in summer and one in winter. I agree that the summer peak is stronger, but it could be argued that the seasonality is more complex than what is highlighted in the main text. Indeed from Fig. 4a you can see warming during winter.	We have fully explained the winter peak of heat transport in your third major comment. Moreover, we added the following sentence to the text. Edit Line 196-199: “Although the peak of heat transport in winter was smaller than that in summer, it was conspicuous (July 2014 and June 2015, Fig. 4b). Such two winter peaks mainly depend on the strengthening of the southward flow and are related to the increase of OSSC due to the strengthening of the wind (Supplementary Fig. 6).”
19	Fig S3: be clear in the caption that here you are showing depth changes of temperature/salinity “contours”.	Done. The caption for the supplementary figure 3 was changed as follows. Edit Line 19-20 in Supplementary information: “Time series variation of the isohaline and isotherm depth with the

		meridional velocity at K4 and Buoyancy flux.”
20	Line159: Conclusions/Discussion/Summary heading is missing.	We agree with Reviewer's opinion. The parts are separated and subtitles are added. Introduction Results - Observation - Seasonal variation of mCDW near the eastern and western flank - mCDW circulation and ocean surface stress curl - Meltwater outflow - Relation between heat transport and meltwater flux Discussion

Reviewer #3.		
ID	Comment Major	Response
1	I see no links to where the mooring data will be available. If not the raw files, the standard within the community these days is at a minimum that data needed to reproduce the figures in the main text. Did I miss this? Are the authors waiting until acceptance?	"Data availability" was omitted from the first submitted manuscript. We added "Data availability" to the resubmitted manuscript and mentioned the source of all data used in this study. In particular, mooring data can check through the Korea Polar Data Center (KPDC), and if someone wants to use the data, they can request it through this website. Then we will consider whether this request is reasonable and decide to share it. Edit Line 374-386: “The mooring data that support the findings of this study are available from the corresponding author upon reasonable request. The detailed information on the data can find on the KPDC (Korea Polar Data Center) website (https://dx.doi.org/doi:10.22663/KOPRI-KPDC-00000636.1) and can request data sharing to the administrator. The model data for wind are obtained from the AMPS website of Ohio State University (http://polarmet.osu.edu/AMPS/) and the NCAR website (https://www.earthsystemgrid.org/dataset/ucar.mmm.amps.html). The sea ice concentration data from AMSR-E, SSMIS and AMSR-2 are obtained from the Sea Ice Remote Sensing website of the University of Bremen (https://seaice.uni-bremen.de/data/). Polar Pathfinder Daily 25 km EASE-Grid Sea Ice Motion Vector, Version 4 data are obtained from NSIDC (National Snow and Ice Data Center) website (https://nsidc.org/data/NSIDC-0116/versions/4). The sea ice thickness data measured from ICESat are obtained from the NASA website (https://earth.gsfc.nasa.gov/cryo/data/antarctic-sea-ice-thickness). The reanalysis model data for net surface heat flux and freshwater flux are obtained from SOSE (Southern Ocean State Estimate) website (http://sose.ucsd.edu).”
	Minor	
2	Line 37-39: “deep-draft ice is more readily accessible” – *more* readily accessible compared to where? Rest of Antarctica? If so, why? Shelf bathymetry? Atmospheric	There seems to be an ambiguous part of the sentence. We added the following sentence by citing previous studies on the conditions (bathymetry) and its mechanism (Ekman transport) for the easy intrusion of warm CDW into the Amundsen Sea continental shelf. In addition, previous studies on the Ross and the Weddell Sea adjacent to the Great Ice Shelf in Antarctica

	forcing?	introduce. Edit Line 37-45: “In the Amundsen Sea (AS), where the seabed depth of the continental shelf is sufficiently deep (>500 m), Circumpolar Deep Water (CDW) intrusion over the slope under the influence of Ekman transport and flow to the south along the deep trough. Therefore, deep-draft ice is readily accessible by warm and salty modified CDW (slightly colder and fresher water compared with CDW; mCDW) and leads to accelerate ice shelf melting and increase mass loss in its glaciers. The oceanic heat transport in the AS is more significant than the Ross and Weddell Seas adjacent to the giant ice shelf in Antarctica. In the Ross and Weddell Seas, where large cyclonic polar gyres are located north of the continental shelf, only a tiny amount of CDW cooled within the gyres intrusion the continental shelf”
3	Line 45: “leading to mass loss” – but there already is mass loss of DIS, so your use of ‘stronger’ implies an ‘increase’ in mass loss.	The introduction has been revised throughout to raise this paper’s importance and provide more information on the study area. The sentence suggesting a correction has been modified and moved as follows. Edit Line 39-41: “Therefore, deep-draft ice is readily accessible by warm and salty modified CDW (slightly colder and fresher water compared with CDW; mCDW) and leads to accelerate ice shelf melting and increase mass loss in its glaciers.”
4	Line 46: “of that circulation” – What circulation is “that” referring to? Across-, or mid-shelf?	‘That circulation’ means the mCDW circulation in front of the DIS that mCDW flows into the cavity and mixed mCDW with meltwater flows out to the ocean. While revising the introduction part, the sentence of the comment was revised as follows. Edit Line 61-63: “In this study, we reveal the unknown seasonal variability of mCDW circulation in front of the DIS and how it propagates into the cavity beneath the ice shelf.”
5	Line 53: “is” to “are”	Done. ‘is’ changed to ‘are’ (see the edit line 69).
6	Line 63-64: “... the current was somewhat weaker” – is this expected?	At the lower layer of the eastern slope, the current showed a strong southward flow, but the current speed became weaker in the upper layer, and the direction turned to the west. These results are expressed as red arrows on the K4 mooring station in the map of Figure 1a. However, we thought the reader might miss this, so we have revised the sentence as follows. Edit Line 80-81: “Away from the seabed, the current was somewhat weaker (near 5 cm s^{-1} at 400 m) and veered westward.”
7	Line 67: “the freezing point” – although stated in the methods, maybe mention here that T_f is function of both S and P.	In this study, the freezing point of seawater for heat content and transport calculate as the sea surface values without considering the pressure. That sentence is modified as follows. Edit Line 84: “Warm and salty mCDW ($2 \square$ higher than the seawater freezing point at the surface)...” Also, the description of Equation (1) in Method change as follows Edit Line 254-255: “.... and T and T_f are seawater temperature and seawater freezing point at the surface based on the salinity, respectively.”

8	Line 69: “relatively cold and freshwater” – change to ‘fresh water’. Also how much colder and fresher? ‘Relative’ is too vague.	In that sentence, "freshwater" was changed to "fresh water", and a specific value adds as follows. Edit Line 86-88: “In contrast, the outflows of relatively cold and fresh water (-1 □, 34.2 PSU) were found above 500 m near the western flank (Fig. 1b and e).”
9	Line 71-72: “depth-independent” – barotropic?	Our current observation data is limited to 400-680 m depth, so it is unreasonable to calculate barotropic flow. Depth-independent flow means an average current from 400 to 680 m.
10	Line 74: did you quantify and compare their periodicity, or is this hand-waving?	From this sentence, we tried to explain that the time-averaged southward flow for each layer decrease toward the mid-depth compared to the lower layer, but its variability (removing the time-average for each layer) does the same in the whole water column. However, "periodicity" is an incorrect descript. We have not looked deeply into its periodicity. Therefore, the sentence has changed as follows. Edit Line 92-93: “The southward component decreased gradually at mid-depth, but its seasonal and intra-seasonal variability remained the same in the whole water column.”
11	Line 77: remove “the”	Done. ‘the remained’ changed to ‘remained’ (see the edit line 95).
12	Line 85: “the salinity difference ...” – what are implications of this? Level of non-divergence? Otherwise, if not important/relevant, consider dropping.	The explanation of the difference in salinity at the eastern slope and the center of the DIS is a necessary sentence for explaining the causes of the increase in the baroclinic component of southward flow with depth. As can be seen in Figure 2b, the difference of the southward flow in the middle layer (400-560 m) is not significant, but it increases toward the lower layer (below 560 m). This is because the density gradient in the zonal direction increases towards the lower layer. In the manuscript submitted for the first time, it may be thought that this sentence is unnecessary because there is not enough explanation about the baroclinic component of southward flow. Therefore, we refined this sentence and added a more detailed explanation as follows. Edit Line 103-107: “The seawater temperature does not differ significantly between the eastern slope and the center, similar to the salinity. Thus, the baroclinic effect by density gradient seems insignificant in the mid-depth layer. This spatial distribution of salinity explains that the vertical shear of the southward flow (Fig. 2b) is not prominent in the middle layer (400-560 m) compared to the lower layer (below 560 m).”
13	Line 91-93: “To estimate ... ” - The start of this paragraph comes out of nowhere. E.g. you haven’t said why OSSC would relate to barotropic variability. Seems like there needs to be an overview/motivation precursor to this “such and such was calculation” sentence.	We fully agree with your opinion. In the previously submitted manuscript, OSSC's calculation results were too sudden. At the request of another reviewer, we first investigated the effect of the variability of buoyancy flux on southward flow due to the melting and formation of sea ice near the DIS. The results were then added to the manuscript (see the edit lines 113-123). However, the results were insufficient to explain the variability of the southward flow. Therefore, before explaining the results of OSSC calculations, we add the following sentence to explain the effect of spatial imbalance of stresses introduced into the ocean due to the sea ice on ocean circulation. Edit Line 124-126: “At the boundary of ASP, the steep gradient of sea ice concentration causes the spatial stress imbalance into the ocean under the homogenous wind field. As a result, it causes the divergence or convergence of the sea surface, and spatial difference of sea surface elevation can be induced

		barotropic current.”
14	Line 106: “southeastward” – consistent hyphenation of these cardinal directions. Sometimes used, sometimes not.	Done. It has been modified to use hyphenation (see the edit line 138 and 149).
15	Line 120: Missing space after period?	Done. There are spaces between sentences (see the edit line 165).
16	Line 125: How does 1% compare to estimates from other ice shelves in Amundsen Sea embayment or elsewhere? It doesn’t sounds like much, so some context will help the reader.	The calculated meltwater fraction in the study compares with other regions (PIG and Shirase Glacier Tongue). Moreover, we added the content as follows. Edit Line 170-171: “This value is similar to PIG (~1.5% at 100-500 m) and Shirase Glacier Tongue in East Antarctica (near 0.8%).”
17	Line 129: “inflow” I expect you mean this to be an inflow into the water column at the mooring, but can also be read as an inflow into the sub ice-shelf cavity.	Yes. "inflow" can be confusing to readers. So we changed it to "discharge" (see the edit line 175).
18	Line 130: “can”? - Do you mean “did”?	The sentence is what we claim in this paper, so 'can' was used. Our observation data can not directly show how increased melt ice emissions contribute to regional sea surface elevation change. However, it is theoretically possible that the increase in the freshwater content due to the discharge of meltwater causes a spatial difference in sea surface elevation and can generate a barotropic current. We changed that sentence to the following. Edit Line 175-177: “A decrease of vertically integrated density along the western side of the ice front by a discharge of meltwater during autumn leads to local sea surface elevation rise and an increase in the northward barotropic current.”
19	Line 131: “plotted” – In a “letters”-style journal with condensed and limited text, avoid sentences describing what was plotted. A more interesting and compelling sentence will describe the process of observation, and the reader can follow the figure citation to follow-along and decide for themselves.	The sentence was changed as follows. Edit Line 177- 178: “The vertical mean meridional current velocity and the upper layer meltwater fraction show a pretty good correlation (Fig. 3e, $r=0.52$, with a 17-day lag).”
20	Line 136: This sentence is awkward, including inconsistent tense. Either “, countervailing” or maybe “... that countervailed”.	Done. ‘countervailed’ changed to ‘countervailing’ (see the edit line 183)
21	Line 137: “Furthermore, ...” – very general, not saying very much. Made worse by use of “could” and ending with “again”. I’d drop it or make it more specific and precise to your findings.	I agree with you. In the text, ‘Furthermore, the northward flows created by the freshening could affect the spread of meltwater again.’ was general. Without further analysis, it might be confusing to readers. The sentence has been deleted.
22	Line 138-140: “Finally, the decrease of meltwater discharge in spring may...” – Reader would benefit from sentence on the basic result here (along with reference	In winter and spring, a decrease in meltwater leads to weak stratification, which makes the easy vertical convection of dense winter water. To help readers understand, follow sentences were added with the reference.

	to the figure), before jumping into its effect.	Edit Line 184-187: “During summer and autumn, the input of glacier meltwater in the Dotson trough strengthens a stratification and prevents vertical convection of dense water ³⁰ . However, the decrease of meltwater discharge in the winter and spring would weaken the stratification and enhances the deep convection of dense winter water.”
23	Homogenize” needs an “s”?	The sentence has been completely revised as follows. Edit Line 187-188: “Homogenized water column by vertical mixing upper layer can render an opportunity for the winter water to descend to the middle layer.”
24	Can also change “may” to “would”, and then adding a citation would be appropriate, perhaps this one?	Done. ‘may’ change to ‘would’ We modify the sentence (see the edit line 184-187) as answered in your 22nd comment and cite some previous studies. Silvano, A. et al. Freshening by glacial meltwater enhances melting of ice shelves and reduces formation of Antarctic Bottom Water. Science adv. 4 , eaap9467 (2018).
25	Line 146: “Not observed in remote sensing...” and “unaccounted for” – really? Consider stereophotogrammetry-based DEM differencing from PGC, recent ICESat-II results, moorings in front of ice shelves, etc., etc. Either be more specific to make it correct, or reign in this (specific) claim of novelty.	In this sentence, we wanted to emphasize that the process by which ocean variability propagates into the ice shelf has not yet been fully elucidated. However, it seems that the previously submitted manuscript may be misleading. Thus we have modified that sentence as follows. Edit Line 200-201: “Such a dramatic seasonal dependence is expected to have significant implications for basal melt. However, it is unknown how the recorded seasonal variability propagates into the ice shelf cavity.”
26	Line 148: remove “that”?	Done. ‘suggests that’ change to ‘suggests’ (see the edit line 202)
27	Line 150-153: “Nevertheless, the ...” – surely it will propagate, but what is the relative size of this effect? Hard to say I’m sure, but the reader may wonder how important this effect will be.	Unfortunately, we cannot quantify the seasonality of heat transport. We obtained the current velocity data in the 400-680m depth, which is about 35% of the total water column at the mooring point (sea bed depth 785m). Therefore, it is impossible to calculate the barotropic component from this data. We only believe that the seasonality of heat transport can propagate into the sub-ice shelf even if the barotropic southward flow does not propagate to the ice shelf because barotropic southward flow delivers the relatively warm and salty water to the south, causing seasonal changes in heat content. We are just talking about the possibility in this sentence.
28	Line 152: drop “s” from “propagates”?	Done. ‘propagates’ change to ‘propagate’ (see the edit line 205)
29	Line 153: drop “s” from “DIS’s”?	Done. ‘DIS’s’ change to ‘DIS’ (see the edit line 206)
30	Line 154-158: Is there a physical reason why a 74 days lag would result in a peak correlation? Does any dynamic theory, or modeling work? Also, I recall some drifting floats advecting under and out of some ice shelves in the region. What was their ‘residence’ time?	Unfortunately, there is no model or dynamic theory that can explain this process. We consider these 74 days lag to be the residence time of mCDW. And it may be depended on the velocity and path of mCDW under the ice shelf, but our data are too limited to discuss the residence time. However, recent research that directly observed the mCDW circulation under the ice shelf using the sea gliders is not very different from our result. We cite this recent paper and add the new sentence as follows. Edit Line 211-213: “As a result of observation of seawater

		circulation under the DIS using the sea gliders during one year from January 2018, the residence time of mCDW was about two months³², which was similar to our result.” Reference: Girton, J. B et al. Buoyancy-adjusting Profiling Floats for Exploration of Heat Transport, Melt Rates, and Mixing in the Ocean Cavities Under Floating Ice Shelves. OCEANS 2019 MTS/IEEE SEATTLE, 1-6, doi: 10.23919/OCEANS40490.2019.8962744 (2019).
31	Also, I'm curious what the correlation looked like for the whole range of lags. Is it a sharp max at 74 days, or more broad peak? This would tell the reader how significant this “74” is.	The cross-correlation result adds to the supplementary figure 7. The relatively high correlation of about 0.5 finds between 70-78 days lag and maximum correlation of 0.505 appear in 74 days lag.
32	Line 163: “relates” – you’ve spent a lot of time arguing causation, so be specific here, or in danger of confusing reader or obfuscating this key (and well supported) finding.	Yes. To show these results more clearly, the existing sentences have been modified as follows. Edit Line 218-221: “The variability of meltwater discharge on the western flank showed a time lag of two months (74-day) from the heat transport on the eastern flank. During autumn, the maximum meltwater outflow was found on the western Dotson ice flank due to the intense summer inflow of warm mCDW along the eastern Dotson ice shelf cavity.”
33	Line 167: “Understanding these processes...” – a throwaway sentence. This can be said about nearly any Earth process. Be more specific (e.g. what exactly needs to be better understood to understand the long-term trend, and for what reason) or dedicate this precious space to something else.	Correct! The previous sentence was too general and not specific. We changed the sentence to the following: Edit Line 227-228: “Therefore, understanding the long-term variability of the atmosphere and the subsequent response of the ocean circulation are likely essential for determining the long-term melting trend.”
34	Line 170: “... previously unknown ...” – this is not an unknown effect (I presume you are referring to the seasonally varying heat transport), thinking of recent work in Ross, Filchner-Ronne, and Amundsen. I am most excited by the west-vs-east differences, the influx-outflow lag, baroclinic-vs-barotropic, and OSSC findings – which would be better to highlight in these closing overview sentences.	Correct! The seasonality of heat transport to the ice shelf is nothing new. Many of these studies have been done on the Antarctic coast, especially in the PIG and the Getz Ice Shelf of the Amundsen Sea Ice Shelf. Based on this fact, we have revised the sentence as follows. Edit Line 228-231: “The circulation pattern of mCDW in the front of the ice shelf and its seasonal variability by atmospheric conditions evidenced here are potentially significant for understanding ice shelf retreat due to climate change and needs further study, e.g., ice-ocean coupled numerical model study considering the ice shelf.”
35	There are some subsection headers, but I still had a hard time keeping track with the purpose of each little bit of text and equations. Perhaps sub-subsections would help.	Done. The parts are separated and subtitles are added. Introduction Results - Observation - Seasonal variation of mCDW near the eastern and western flank - mCDW circulation and ocean surface stress curl - Meltwater outflow

		- Relation between heat transport and meltwater flux Discussion
36	Line 197: regarding method of obtaining velocities below 600m at K4 – will seasonality bias your method at all?	When the southward flow is divided into barotropic and baroclinic components, our calculation mainly reflects the variability of the barotropic component, and the baroclinic component reflects only the averaged differences between each layer evaluated from observation data for the previous year. Therefore, as you comment, the seasonal bias of the baroclinic component can be increased the error. However, it was confirmed that the difference in the baroclinic component between each layer was almost constant in the observation data for the previous year. We have added the following sentences to give the reader accurate information about our calculation. Edit Line 258-260: “In this calculation process, it was assumed that the shear between each layer of the baroclinic component of southward velocity was always constant.”
37	Also, Lines 200-203 – these regression equations didn’t do much for me. Are they needed? I think it’s enough to show Supp Figure 4.	Done. I deleted the equation. Supplementary figure 4 was changed Supplementary figure 8.
38	Line 215: “especially” – don’t know why this word is used.	Done. I deleted ‘especially’.
39	Line 223: “low layer” – ambiguous - what depths does this refer to?	Correct! The sentence "at the low layer" is changed to "above the mCDW layer" (see the edit line 280-281).
40	Line 257: “if U_{ice} assumed to be far larger than U_w ” – is this a good assumption? Not good enough for me to not include a citation or other evidence.	Correct! In a typical ocean, the current velocity can not negligible because it is not tiny compared to the sea ice velocity. In addition, in equation (21) for calculating the OSSC, the current velocity was not neglected. However, to solve the free drift sea ice motion of equation (6), we assume an ideally stable ocean with a current velocity of 0 and simplify the second and third terms of equation (6) to calculate the drag coefficients boundary between air and sea ice and between sea ice and ocean. These calculation methods are suggested by Lu et al. (2016). The sentence has been modified as follows. Edit Line 325-327: “In an idealized steady ocean, the second and third terms in equation (6) can be rewritten as $-R(\theta_w)\rho_w C_{D,io} U_{ice} (U_{ice})$ and $-R(\pi/2)\rho_{ice}hfU_{ice}$, assuming that the current velocity is close to zero.”
41	Line 276: h_{ice} is taken from ICESat – does the interannual variability of h_{ice} preclude you from using this method?	To calculate the horizontal distribution of drag coefficient between sea ice and ocean from 2006 to 2018, the sea ice thickness data in the study area used the average of all data observed in the spring and autumn from 2004 to 2007 from ICESat. Unfortunately, we had no observed sea ice thickness data in 2014 and 2015 by satellite. During this period, the Ice Bridge Sea Ice Campaigns observed the sea ice thickness, but data was insufficient. We did not consider the temporal variation of drag coefficients to calculate OSSC in the study area, but only spatial distributions. The drag coefficient between air-sea ice and sea ice-ocean is calculated by the response of sea ice motion (wind factor, turning angle) under the wind field for a long period of more than ten years. At this time, the sea ice thickness is constant, and only spatial variability is considered. Therefore, if the sea ice thickness has a long-term trend, the error of the

		estimated drag coefficient can increase due to the difference of observed periods between sea ice thickness and other variables. We think it is a comment about the error that may occur due to the close-to-decade difference between the OSSC calculation period and the sea ice thickness observation period. According to a recent study of the decadal variation of sea ice thickness in the Amundsen Sea (Wang et al., 2020), the average sea ice thickness had an interannual variation. However, it had no considerable difference between average sea ice thickness from 2004 to 2007 and 2014. Also, they do not show a significant long-term trend from 2003 to 2017. Therefore, the error due to the difference in the observation period of sea ice thickness is negligible. Reference Wang, X., Jiang, W., Xie, H., Ackley, S., & Li, H. (2020). Decadal Variations of Sea Ice Thickness in the Amundsen-Bellingshausen and Weddell Seas Retrieved From ICESat and IceBridge Laser Altimetry, 2003–2017. Journal of Geophysical Research: Oceans, 125(7), e2020JC016077.
42	Figure 1 and throughout: do you need to label “salinity” with units?	Done. We added ‘PSU’
43	Figure 1a – is the contouring of overview map ocean depths, or ocean temperatures (it appears unlabeled)? Seems a difference color map than other contours.	The contour of the map is the ocean depth. The legend has been modified to make it identifiable.
44	Figure 2: although K4 is mentioned, remind readers whether it is on East or West flank.	Done. ‘Eastern flank’ was added
45	Supp Figure 3 caption: Leave out the “show a good correlation”, if indeed a correlation was actually done – either way, this is a result and should be in the text. The figure captions are for describing what is plotted only.	Done. In the caption of Supplementary Figure 3, "Salinity (34.2 PSU, red) and potential temperature (-1.4 °C, red) show a good correlation with meridional velocity." has been deleted. (see the edit line 23 in Supplementary information).
46	Supp Figure 4b: The HF variability in this time series is so high that I’m not sure what is supposed to be communicated to the reader. LP filter? Zoom in on some period?	In order to remove the short fluctuation, we plotted the 9 day moving average of the results. And Supplementary Figure 4 was changed Supplementary Figure 8.
47	Supp Figure 5: c and d panels don’t tell me much. I’d recommend dropping if not in Supplement.	Done. Supplementary Figure 5 was changed Supplementary Figure 4. Supplementary Figure 4. c and d have been removed.

Reviewers' Comments:

Reviewer #1:

Remarks to the Author:

I thank the authors for their clear response to my suggestions. My only remaining comment is a repeat from last time. I think the Discussion fall short in contextualising the Dotson with what we know about other systems. There are snippets of this through the manuscript but the Discussion would be a key time to synthesize these points.

Reviewer #2:

Remarks to the Author:

As highlighted in the previous review, the dataset collected by Yang et al is exceptional. Two years of observations in front of an Antarctic ice shelf are extremely rare and can provide a strong improvement in understanding ice ocean interaction and sea level rise. The seasonal variability of ocean heat transport into the ice shelf cavity and basal meltwater outflow are well described. Increased heat transport is observed in summer and increased meltwater outflow is observed a couple of months afterwards. I agree with the authors that the surface stress curl plays a primary role in this, but I believe some further clarification is required in the manuscript. This is highlighted below in the major comments. I also have few minor comments.

Major comments

- Line 121-123: the buoyancy flux appears to covary on seasonal time scale with the meridional flow (figure S3). Please clarify how this plot shows that buoyancy forcing is not important for seasonal variability.
- Line 129-135: The cross correlation shows both positive and negative correlations, depending on the lag. How do you interpret this? Consider that if you apply a filter, the lag-correlation/coherence needs to account for that.

Minor Comments

- Introduction: I like the way the introduction is structured. However, some re-wording is required to improve readability.
- Figure 2d: specify in the caption that one of the axis is reversed.
- Line 516: "Average potential temperature and salinity near the top (270m) and bottom layer (745m) are highlighted".
- Line 95-96: Please rephrase the sentence to clarify.
- Line 98: "affect the seasonal variability of the southward flow."
- Line 114: "can affect".
- Line 155-118: positive buoyancy flux implies density reduction, negative buoyancy is the opposite. Please be consistent with this in the text.
- Line 121-123: the surface buoyancy flux is estimated using a model. Please specify this.
- Line 126: "can induce".
- Line 141: "strengthening".
- Line 187: "vertical mixing in the upper layer can provide an opportunity...".

Reviewer #3:

Remarks to the Author:

Overall comments:

For the most part, the authors successfully addressed most of the reviewers' concerns, comments, and suggestions. However, while the quality of the intellectual component of the revised manuscript is much improved, the added text (apart from punctuation or single word edits) suffers from severe grammar deficiencies, rendering the readability much lower than the first draft. This wasn't the case in the first version I read, so maybe there wasn't sufficient proofreading for this version. Unfortunately, these grammar and readability edits make up the bulk of my comments in this review, leaving me with the feeling that I spent most of the time on copy-editing when there are interesting science findings that failed to stick because of communication shortfalls. Since the early draft didn't have these problems, I'm confident that the author has the ability and team to fix this fairly critical issue in short order.

There is certainly better motivation in this version of the manuscript, but still leaves me wanting to know some more specifics about the value of knowing interannual and intra-annual variations. Is there a way you can quantify this value? For example, how much would you (or have previous studies) over or underestimated *annual* heat flux with limited hydrographic data? I imagine if one extrapolated December-January only measurements to be annual values, there would be fairly different MegaWatt estimates. This could then be used to motivate mooring or other year-round measurements.

Another general comment is that perhaps the paper could benefit from a schematic/cartoon. I drew three different 2D cross-section across the ice front, with arrows and isopycnals. This made me realize that one or two simple schematics of the relevant process and feedbacks would go a long way in helping the reader follow along (this became especially clear in the sea ice/atm forcing sections). I know you are pressed for space in the main text, but take advantage of the unlimited Supp Mat.

Comments on "Response to Reviewer #1"

- For motivation, consider how much inland ice Smith, Pope, and Kohler, drain? What is the Sea Level Equivalent?

- Also brings up the potential bias in heat flux that summer-only studies would have. Can you quantify/estimate this bias? I'd love to know what XX% those type of studies are biased.

- Comment 7 (with profile figure) – I really like this CTD profile, as it really illustrates what you state in the text, and even reinforces the likelihood of missing meltwater above the topmost mooring. Add this to paper would be really nice, even if just in Supp Mat.

Minor Comments:

Line 41: grammar - "accelerate" to "accelerated"?

Line 43: grammar - "shelf" to "shelves"

Line 44-45: grammar – "cooled within the gyres intrusion the continental shelf". First, is "gyres" possessive? Second, maybe need to make it "on the continental shelf"

Line 48: Even though the reader CAN follow the citations, I am always disappointed with introduction sentence that just plainly state "So and so was previously studied." The reader gets much more out of an edited sentence that says "...was previously studies and found this and that."

Line 50: grammar – Why is "Polynya" capitalized? Also, consider dropping "the" before "sea ice".

Line 53: grammar – "Such elevated rates compared to..."?

Line 55: grammar – Needs to be either "Previous" or "Previously," (add comma)

Line 58: grammar – "has mainly". "Was"?

Line 59-61: this is where I'd like to hear more specifics about how knowing these seasonal variations can help. It's nice you said that this new knowledge will help, but the reader wants to know how (even broadly, or speculatively)

Line 105:- consider "suggests" instead of "explains". Also, grammar of second half of sentence around "prominent" is a little confusing. Maybe something like "... less prominent ..."

Line 113: grammar - "In the north of DIS,..." to something like "North of the DIS, the ASP..."

Line 114: grammar - "can be affected seawater" to something like "can affect seawater"

Line 115-117: am I mistaken, or are your negative and positive buoyancy's reversed? Or is this a different convention? If so, please state/clarify.

Line 122: grammar - adjust to leave out the semicolon, as it hinders readability of this sentence. Also consider rewording to something like "... they do not agree well, implying that the effect of buoyancy..."

Line 124: add "the" before ASP?

Line 125: maybe change to "... under a homogeneous wind field"

Line 126: grammar - maybe change to "... and resulting spatial difference ...". Also need to edit the "can be induced barotropic current."

Line 141: grammar - "... by causing the strengthen of southward ..." to something like "... by strengthening southward mCDW flow."

Line 164: Point us to the equation (and/or relevant cite) here.

Line 189: Consider using the more common ACC for Antarctic Circumpolar Current (instead of your AACC).

Line 197: grammar - "conspicuous" - I like such colorful wording in science articles! But I'm not sure what is meant by this here. Elaborate? Or tie to previous or subsequent sentence?

Line 229-231: Similar to minor comment above, WHY is seasonal variability "potentially significant for understanding ice shelf retreat". Saying that it is significant is step one for motivating this paper, but need to know why. What future studies/efforts will benefit by and build on your paper here?

Line 286: grammar - the continued capitalization of "Polynya" confuses me. Also, this flux is not specific to polynyas, but a result of sea ice formation more generally, right?

Line 294: grammar - "obtained"

Line 513: b & c - I like these vector diagrams more than in the previous draft, but consider adding a symbol to highlight "the start" of each (I know it's (0,0) but better to be clear)

Figure 3e - The new labeling of this figure helps a lot. But makes me wonder if there's a more you can do to add evidence of this relationship. Graphically, perhaps you could shade north and south meridional velocities? If you "de-seasonalized" the time series (even a simple centered moving average) this shading would be even more impactful. Quantitatively, consider normalized cross correlation.

Response to reviews of “Seasonal variability of ocean circulation near the Dotson Ice Shelf, Antarctica”

Manuscript NCOMMS-21-02666A

Response to Reviewers

We would like to thank the reviewers and editors for their time and effort in reviewing the paper "Seasonal Variability in Ocean Circulation Near the Dotson Ice Shelf, Antarctica," submitted for publication in Nature Communications. Detailed responses to the comments and explanations how the manuscript was changed in response to the comments are provided below. Line numbers refer to the new version of the manuscript. All changes are highlighted in the manuscript. See step-by-step answers to reviewers' comments below. Here, all line numbers refer to revised manuscript files.

Reviewer #1.		
ID	Comment Major	Response
1	I thank the authors for their clear response to my suggestions. My only remaining comment is a repeat from last time. I think the Discussion fall short in contextualising the Dotson with what we know about other systems. There are snippets of this through the manuscript but the Discussion would be a key time to synthesize these points.	In previous versions of the paper, the discussion section was similar to the conclusion section summarizing the study. We thank the reviewer for pointing this out, and apologize for somewhat misinterpreting the previous comment. We have rewritten and rearranged the Discussion section and now point out the differences and similarities with other ice shelf systems. The revised Discussion mainly covered the following. 1. The large scale circulation, which as far as we can tell is similar in many aspects to almost all ice shelf systems (Edit Line 239-249)2. Quantitative evaluation and seasonal variability of heat transport into the DIS along the eastern slope and meltwater flux along the western slope (Edit Line 250-266).3. The mechanism of heat transport in other ice shelves in the Amundsen Sea (Getz Ice Shelf, Pine Island Glacier) and the mechanism of mCDW inflow into the DGT connected to the DIS, consistent with the results of this study that are mainly dependent on wind, and the long and short-term variability of the atmospheric circulation (Edit Line 267-288).4. Summary of this study and the necessity for additional research (Edit Line 289-302).

Reviewer #2.

ID	Comment Major	Response
1	Line 121-123: the buoyancy flux appears to covary on seasonal time scale with the meridional flow (figure S3). Please clarify how this plot shows that buoyancy forcing is not important for seasonal variability.	We have expanded this section in order to make the reasoning clearer. The reason for analyzing the buoyancy flux and OSSC was to identify the factors affecting the seasonal variation of mCDW flowing into the ice shelf. Variability of sea ice concentration in the polynya causes seasonal variability in ocean surface buoyancy flux but also variability in ocean surface stress curl. Although the southward flow affected by OSSC can show a similar seasonal variability as the buoyancy flux, it cannot be concluded that the buoyancy flux affects the southward flow. To examine the effect of the volatility of buoyancy flux on the southward flow, we examined their relationship after removing the seasonal cycles of the meridional velocity and buoyancy flux. With the seasonal cycle removed (90-day high-pass filter), the correlation between them was 0.06, which was statistically insignificant in the 99% confidence interval. These results imply that the buoyancy flux and the southward flow are both affected by atmospheric variability on the seasonal time scale, but the influence of the buoyant flux on the seasonal variability of the southward flow is insignificant.  Figure. Correlation between Buoyancy flux and meridional velocity. We added the following text in order to make this more clear in result part. Edit Line 133-139: “Both local buoyancy flux and meridional velocity demonstrate a seasonal variation that decreased in summer and increased in winter. However, after removing the seasonality from both time series there was no statistically significant correlation between local surface buoyancy flux and meridional velocity. Therefore, although both buoyancy flux and meridional velocity are influenced by atmospheric variability and show a good agreement in seasonal time scale, the effect of local buoyancy fluxes on

		mCDW variability and circulation are comparatively weak in the present data set.”
. 2	Line 129-135: The cross correlation shows both positive and negative correlations, depending on the lag. How do you interpret this? Consider that if you apply a filter, the lag-correlation/coherence needs to account for that.	In Fig. 2e, the OSSC and meridional velocity mostly showed a negative correlation, indicating that the southward flow became stronger as the OSSC increased. On the other hand, it shows a weak positive correlation ($r=0.14$) near 0 lag, so it can be misunderstood that an increase in OSSC causes a northward flow. However, in the U-shaped trough where the south, east, and west sides are blocked, a southward flow occurs on the surface layer and a northward flow on the bottom layer by the easterly wind, as shown in figure 2. In the figure, the negative correlation between the zonal wind and the meridional velocity means that there is a positive correlation between the easterly wind and the northward flow. In addition, the strengthening of the easterly wind generally increases the shear of the easterly wind in the meridional direction and increases the OSSC, so there may be a positive correlation with the northward flow near the bottom, as below figure 1. In the figure, the negative correlation between the zonal wind and OSSC means that there is a positive correlation between the easterly wind and OSSC. This positive correlation between depth-averaged velocity and easterly wind and OSSC is because our mooring data is biased toward the bottom. Conversely, if our observed data are biased upwards, then the depth-averaged velocity may have a negative correlation with the east wind and OSSC. On the other hand, in the cross-correlation result between OSSC and meridional velocity, the sign of correlation according to lag does not change by applying a filter as below figure 3. However, by removing short frequencies, the correlation between them is amplified. As a result, the cross-correlation was changed to a significant correlation after applying the filter at the insignificant lag, but the 28 lag showed a significant correlation at the 99% confidence interval regardless of the filter application.

Figure 1. Cross correlation between zonal wind and Ocean surface stress curl. $r=-0.47$ at '0' lag.

Figure 2. Cross correlation between zonal wind and meridional velocity.

Figure 3. Cross correlation between unfiltered OSSC and depth averaged meridional velocity.

	Minor	
3	Introduction: I like the way the introduction is structured. However, some re-wording is required to improve readability.	In order to improve readability, insufficient English expression and grammatical errors have been corrected throughout the manuscript, including the introduction.
4	Figure 2d: specify in the caption that one of the axis is reversed.	Done. The following sentence was added. Edit Line 630: "Axis of the OSSC was reversed."
5	Line 516: "Average potential temperature and salinity near the top (270m) and bottom layer (745m) are highlighted".	Done. The sentence was changed as follows: Edit Line 619-620: "Average potential temperature and salinity near the top (270m) and bottom layer (745m) are highlighted"
6	Line 95-96: Please rephrase the sentence to clarify.	Original sentence was "Vertical meridional current shear increased toward the bottom remained nearly constant during the measured period." This sentence has been modified as follows. Edit Line 103-105 : "The vertical shear of the meridional current component was largest in the warm layer near the bottom, and it had comparatively small temporal variations through the measured period."
7	Line 98: "affect the seasonal variability of the southward flow."	Done. The sentence was changed as follows: Edit Line 107: "...affect the seasonal variability of the southward flow."
8	Line 114: "can affect".	Done. 'be' was deleted from 'can be affect. (See the edit line 123)
9	Line 115-118: positive buoyancy flux implies density reduction, negative buoyancy is the opposite. Please be consistent with this in the text.	Regarding buoyancy flux at the air-ocean interface, some studies express the inflow of heat and freshwater into the ocean as a positive buoyancy flux based on the ocean, while some studies express this as a negative buoyancy flux based on the atmosphere. We followed the latter. However, as the reviewers pointed out, it can lead to confusion. Therefore, we supplemented the sentence as follows to minimize confusion. Edit Line 124-129: "Previous study from PIG ²² suggests that local positive (i.e., from ocean to air) buoyancy fluxes from sea ice formation and/or atmospheric cooling (less buoyant at the sea surface) creates deep convection, leading to a downward descent of the thermocline and thinning of the mCDW layer at the bottom, while negative buoyancy fluxes (i.e., downward and more buoyant at the sea surface) due to surface heating and sea ice melting leads to an upward movement of the thermocline and a increase mCDW volume."
10	Line 121-123: the surface buoyancy flux is estimated using a model. Please specify this.	Done. The sentence was changed as follows. Edit line 130-131: "...local surface buoyancy fluxes were calculated using the heat- and freshwater fluxes from the data-assimilating Southern Ocean model."

11	Line 126: “can induce”.	Done. ‘can be induced’ change to ‘can induce’. (See the edit line 143)
12	Line 141: “strengthening”.	Because we have modified the sentence as follows, existing ‘strengthen’ is changed to ‘strengthened’. Edit Line 157-158: “The summertime increase of the OSSC leads to a strengthened southward mCDW flow (Supplementary Fig. 6a) and enhanced heat transport to the ice shelf”
13	Line 187: “vertical mixing in the upper layer can provide an opportunity...”.	Done. The sentence was changed as follows. Edit Line 206-207: “Homogenized water column by vertical mixing in the upper layer can provide an opportunity for the winter water to descend to the middle layer.”

Reviewer #3.		
ID	Comment Major	Response
1	However, while the quality of the intellectual component of the revised manuscript is much improved, the added text (apart from punctuation or single word edits) suffers from severe grammar deficiencies, rendering the readability much lower than the first draft.	In order to improve readability, insufficient English expression and grammatical errors have been corrected throughout the manuscript.
2	There is certainly better motivation in this version of the manuscript, but still leaves me wanting to know some more specifics about the value of knowing interannual and intra-annual variations. Is there a way you can quantify this value? For example, how much would you (or have previous studies) over or underestimated *annual* heat flux with limited hydrographic data? I imagine if one extrapolated December-January only measurements to be annual values, there would be fairly different MegaWatt estimates. This could then be used to motivate mooring or other year-round	Heat transport and meltwater flux calculated from observation data limited to the existing summer season did not consider their seasonal variability, so quantitative evaluation of the annual average was impossible. The new observations we present here can be evaluated quantitatively to that, which is an important point that further enhances the value of this study, as your comment. We evaluated how much the summer average differs from the annual average from the seasonal variability of heat transport and meltwater flux to and from the ice shelf. In addition, the importance of these results and the need for further research were mentioned as follows. Edit Line 254-263: “The substantial seasonal variability of heat transport and meltwater discharge near the ice shelf calving front that was observed, suggests that previously evaluated heat- and meltwater transport ^{15,24,25} , based on summer observation, may have been overestimated. Based on the present data set the average heat transport in summer was 141 MW m ⁻¹ , 1.27 times

	measurements.	greater than the annual average of 111 MW m^{-1} , a number that can be used to scale summertime observations from this region. The seasonal average meltwater flux to the north was at a maximum of $89.2 \text{ cm}^2 \text{ s}^{-1}$ in autumn and $25.7 \text{ cm}^2 \text{ s}^{-1}$ in summer. In contrast, meltwater flux showed negative in winter and spring due to the dominant southward flows, and the annual average was close to zero. This seasonality implies that additional observations are needed to determine how much of the inter-annual variability of heat and meltwater transport seasonality propagates into the ice shelf cavity.”
3	Another general comment is that perhaps the paper could benefit from a schematic/cartoon. I drew three different 2D cross-section across the ice front, with arrows and isopycnals. This made me realize that one or two simple schematics of the relevant process and feedbacks would go a long way in helping the reader follow along (this became especially clear in the sea ice/atm forcing sections). I know you are pressed for space in the main text, but take advantage of the unlimited Supp Mat.	One of the important point in this paper is that the interaction between the ocean and atmospheric ice produces seasonal fluctuations in the ocean circulation, which in turn influences the inflow of mCDW and the discharge of melted ice. I agree that a schematic diagram will be helpful to understand this process. Therefore, a schematic diagram to explain this process (OSSC-sea surface-barotropic southward current) has been added to supplementary figure 6. 	Comments on “Response to Reviewer #1”	
4	For motivation, consider how much inland ice Smith, Pope, and Kohler, drain? What is the Sea Level Equivalent?	Thanks for the additional comments. Although we did not find a previous study that calculation of a sea-level equivalent by only Kohler and Smith Glaciers buttressed by the Dotson Ice Shelf (DIS), however, the previous studies (Mouginot et al., 2014) suggest that global sea-level rise by about 1.2m if all of the major glaciers in the Amundsen Sea Embayment melted completely. In addition, his study showing that ice drain by Pope, Smith, and Kohler Glaciers have increased more recently than other Glaciers in the ASE. We add the following sentence in the introduction section. Edit Lien 38-46: “At the Amundsen Sea Embayment (ASE), ice from the West Antarctic Ice Sheet (WAIS) is drained into the ocean through the Pine Island, Thwaites, Haynes, Smith, Pope, and Kohler glaciers. These glaciers had an ice flux of $334 \pm 15 \text{ Gt yr}^{-1}$ in 2013⁹, and they have potential to impact sea level rise globally should this flux change significantly. In the 1980s, Pope, Smith, and Kohler Glaciers, located in the western ASE, drained ice about 15% of the total ice mass loss from the ASE, but

		since 2013 their rate have increased and they now contribute about 23%. In the ASE, warm and salty Circumpolar Deep Water (CDW) can intrude from the deep ocean across the continental shelf under the influence of wind and Earth’s rotation ¹⁰ .”
5	Also brings up the potential bias in heat flux that summer-only studies would have. Can you quantify/estimate this bias? I’d love to know what XX% those type of studies are biased.	As mentioned in the previous major comment, the summer average heat transport is 1.27 times larger than the annual average. Thus, the estimated heat transport from the only summer observed data might be overestimated by about 27% of the annual average. As to answer comment 2, we have added some sentences to the discussion part as follows. Edit Line 256-259: “Based on the present data set the average heat transport in summer was 141 MW m^{-1} , 1.27 times greater than the annual average of 111 MW m^{-1} , a number that can be used to scale summertime observations from this region.”
6	Comment 7 (with profile figure) – I really like this CTD profile, as it really illustrates what you state in the text, and even reinforces the likelihood of missing meltwater above the topmost mooring. Add this to paper would be really nice, even if just in Supp Mat.	Because we mentioned the limitation of calculated meltwater fraction from mooring data in the main text that does not cover the upper layer, we agree to add a profile of the meltwater fraction calculated from shipboard CTD as you suggest in supplementary. The figure below has been added to supplementary.  Supplementary Figure 7 Calculated meltwater fraction from the shipboard CTD data (temperature, salinity, dissolved oxygen) measure in January 2014 and 2016 at the western mooring station (K5). The blue box indicates the covered water column by K5 mooring.
	Minor	
7	Line 41: grammar - “accelerate” to “accelerated”?	Because we have modified the sentence as follows, existing ‘accelerate’ is changed to ‘accelerates’.

		Edit Line 47-49: "... and in the southern end of these troughs a modified version of CDW (slightly colder and fresher water compared with CDW; mCDW) can access deep-draft ice which accelerates the ice shelf melt and the glacier mass loss ¹¹⁻¹⁷ ."
8	Line 43: grammar - "shelf" to "shelves"	Done. 'shelf' change to 'shelves'. (See the edit line 51)
9	Line 44-45: grammar - "cooled within the gyres intrusion the continental shelf". First, is "gyres" possessive? Second, maybe need to make it "on the continental shelf"	Done. First, 'Gyres' change to 'gyre'. Second, 'on' was added. (See the edit line 52)
10	Line 48: Even though the reader CAN follow the citations, I am always disappointed with introduction sentence that just plainly state "So and so was previously studied." The reader gets much more out of an edited sentence that says "...was previously studies and found this and that."	Done. The sentence was changed as follows. Edit Line 54-58: "Recently, it has been found that the surface water flowing into the cavity influences Ross and Filcher-Ronne ice shelves melting and their seasonality ^{20,21} . Long-term mooring observation in front of Pine Island Glacier (PIG) have shown that the variation in sea surface heat flux can affect the variability in CDW volume and seawater circulation pattern ²² ."
11	Line 50: grammar - Why is "Polynya" capitalized? Also, consider dropping "the" before "sea ice".	Done. We corrected all capital letters in polynya in the text. And 'the' was removed. (See the edit line 59)
12	Line 53: grammar - "Such elevated rates compared to...?"	Done. This means that the ice thickness change in DIS is faster than in the Amundsen Sea sector. Added specific value and the sentence was changed as follows. Edit line 61-63: "... in the ASE, buttresses the Kohler and Smith Glaciers and has thinned by 2.6 m yr ⁻¹ between 1994 and 2012, which is 30% faster than the average thickness change in the AS sector (1.94 m yr ⁻¹) ⁵ ."
13	Line 55: grammar - Needs to be either "Previous" or "Previously," (add comma)	The 'previous' has been deleted because the sentence has been modified as follows. Edit Line 66-68: "Although there is a clear seasonal variation of the mCDW intrusions further north in the Getz-Dotson trough ^{10,23} , previous records at the ice shelf front ^{15,24,25} have been too short to determine any seasonality there."
14	Line 58: grammar - "has mainly". "Was"?	'has mainly' was deleted as the sentences were corrected same as comment #13.
15	Line 59-61: this is where I'd like to hear more specifics about how knowing these seasonal variations can help. It's nice you said that this new knowledge will help, but the	We agree that more specific comments are needed on the importance and contribution of seasonality in the mCDW circulation. We add some sentence in introduction part. Edit Line 68-72: "The seasonal variation of atmospheric forcing into the ocean caused by the seasonality of the sea

	reader wants to know how (even broadly, or speculatively)	ice distribution will affect the ocean circulation near the ice shelf ²⁶ . Therefore, confirming the seasonality of the mCDW circulation in front of the ice shelf and identifying the causes is essential for quantifying the influx of oceanic heat into the ice shelf cavity and understanding the thinning process of the ice shelf.”
16	Line 105:– consider “suggests” instead of “explains”. Also, grammar of second half of sentence around “prominent” is a little confusing. Maybe something like “... less prominent ...”	Done 1. ‘explains’ change to ‘suggests’. (See the edit line 115) 2. ‘not prominent’ change to ‘less prominent’. (See the edit line 115)
17	Line 113: grammar – “In the north of DIS,...” to something like “North of the DIS, the ASP...”	Done. ‘In the north of DIS, Amundsen sea Polynya’ change to ‘North of the DIS, the Amundsen...’. (See the edit line 122)
18	Line 114: grammar – “can be affected seawater” to something like “can affect seawater”	Done. ‘can be affected seawater’ change to ‘can affect seawater’. (See the edit line 123)
19	Line 115-117: am I mistaken, or are your negative and positive buoyancy’s reversed? Or is this a different convention? If so, please state/clarify.	Regarding buoyancy flux at the air-ocean interface, some studies express the inflow of heat and freshwater into the ocean as a positive buoyancy flux based on the ocean, while some studies express this as a negative buoyancy flux based on the atmosphere. We followed the latter. However, as the reviewers pointed out, it can lead to confusion. Therefore, we supplemented the sentence as follows to minimize confusion. Edit Line 124-129: “Previous study from PIG ²² suggests that local positive (i.e., from ocean to air) buoyancy fluxes from sea ice formation and/or atmospheric cooling (less buoyant at the sea surface) creates deep convection, leading to a downward descent of the thermocline and thinning of the mCDW layer at the bottom, while negative buoyancy fluxes (i.e., downward and more buoyant at the sea surface) due to surface heating and sea ice melting leads to an upward movement of the thermocline and a increase mCDW volume.”
20	Line 122: grammar – adjust to leave out the semicolon, as it hinders readability of this sentence. Also consider rewording to something like “... they do not agree well, implying that the effect of buoyancy...”	We write sentences to answer other reviewer's major comments “the buoyancy flux appears to covary on seasonal time scale with the meridional flow (figure S3). Please clarify how this plot shows that buoyancy forcing is not important for seasonal variability.”. The sentence that pointed out the grammatical error has been corrected as follows. Edit Line 134-139: “However, after removing the seasonality from both time series (by 3-month high-pass filtering of the data) there was no statistically significant correlation between local surface buoyancy flux and meridional velocity. Therefore, although both buoyancy flux and meridional velocity are influenced by

		atmospheric variability and vary seasonally, the effect of local buoyancy fluxes on mCDW variability and circulation are comparatively weak in the present data set”
21	Line 124: add “the” before ASP?	Done. ‘the’ was added. (See the edit line 140)
22	Line 125: maybe change to “... under a homogeneous wind field”	Done. ‘the’ change to ‘a’. (See the edit line 141) And existing sentence was change to as follows. Edit Lin 141 : “When a homogeneous wind field blows over an opening in the fast ice,...”
23	Line 126: grammar – maybe change to “... and resulting spatial difference ...”. Also need to edit the “can be induced barotropic current.”	Done. ‘resulting’ was added. (see the edit line 142) The sentence was changed as follows. Edit Line 141-143: “When a homogeneous wind field blows over an opening in the fast ice, spatial stress gradients (Ocean Surface Stress Curl, OSSC, Fig. 2c) are created near the edges, resulting in divergence or convergence of the wind-driven surface (Ekman transport) ²³ . These can induce barotropic currents. ”
24	Line 141: grammar – “... by causing the strengthen of southward ...” to something like “... by strengthening southward mCDW flow.”	Done. The sentence was changed as follows. Edit line 157-158 : The summertime increase of the OSSC leads to an strengthened southward mCDW flow (Supplementary Fig. 6a) and enhanced heat transport to the ice shelf.”
25	Line 164: Point us to the equation (and/or relevant cite) here.	Done. The sentence was changed as follows. Edit line 182-183 : “...maximums of the meltwater fraction (e. g. concentration of glacial meltwater in seawater calculated by eq. 3 and 4 in method) (Fig. 3b and d).”
26	Line 189: Consider using the more common ACC for Antarctic Circumpolar Current (instead of your AACC).	In general, 'ACC' is an abbreviation for 'Antarctic Circumpolar Current'. And in some previous studies, 'AACC' or 'ACoC' was used as an abbreviation for 'Antarctic Coastal Current' (Kim et al., 2016). However, in this manuscript, Antarctic Coastal Current is no longer mentioned in other sentences except for this sentence, so there is no need to use the abbreviation, so it has been changed as follows. Edit line 208: “...convection and the Antarctic Coastal Current ^{26,28} to the westward can rapidly reduce upper layer...”
27	Line 197: grammar – “conspicuous” – I like such colorful wording in science articles! But I’m not sure what is meant by this here. Elaborate? Or tie to previous	Although the winter peak is smaller than the summer peak, it stands out compared to the small peaks in spring and autumn. We have corrected the previous incomplete sentence as follows. Edit line 216-217: “Although the peak of heat transport

	or subsequent sentence?	in winter was smaller than that in summer, it was conspicuous (July 2014 and June 2015, Fig. 4b) compared to the small peaks in spring and autumn.”
28	Line 229-231: Similar to minor comment above, WHY is seasonal variability “potentially significant for understanding ice shelf retreat”. Saying that it is significant is step one for motivating this paper, but need to know why. What future studies/efforts will benefit by and build on your paper here?	We made the ambiguous sentence more concrete by amending the last paragraph of the manuscript as follows. It also emphasized the importance of the conclusions found in the results of this study and the need for further research. Edit Line 289-302: “The circulation pattern of mCDW in the front of the ice shelf and its seasonal variability by atmospheric conditions demonstrated here allow for quantitative evaluation of the effects of short-term variability in the atmosphere on ocean circulation. This finding implies that the ocean circulation effect by local meteorological conditions in front of the ice shelf plays an essential role in regulating mCDW inflow and the basal melting of the ice shelf. Therefore, understanding the long-term variability of the atmosphere and the subsequent response of the ocean circulation are likely essential for determining the long-term melting trend of the WAIS. Furthermore, these results will improve our understanding of the effects of long-term variability in the atmosphere, such as climate change, on ice-shelf retreat, and we will be more confident about the need for further studies, e.g., ice-ocean coupled numerical model study considering the ice shelf.”
29	Line 286: grammar – the continued capitalization of “Polynya” confuses me. Also, this flux is not specific to polynyas, but a result of sea ice formation more generally, right?	You're right. This equation is commonly applied to the ocean. Therefore, ‘in polynya’ has been deleted. (See the edit line 358)
30	Line 294: grammar – “obtained”	Done. ‘obtain’ change to ‘obtained’ (See the edit line 365)
31	Line 513: b & c – I like these vector diagrams more than in the previous draft, but consider adding a symbol to highlight “the start” of each (I know it’s (0,0) but better to be clear)	Done. Added a symbol to the start point. (See the Figure 1 b and c)
32	Figure 3e – The new labeling of this figure helps a lot. But makes me wonder if there’s a more you can do to add evidence of this relationship. Graphically, perhaps you could shade north and south meridional velocities? If you “de-seasonalized” the time series (even a simple centered moving average) this shading would be even more impactful.	We conducted several experiments to solidify further our conclusion that the variability of the meltwater fraction on the western slope of the DIS affects the meridional velocity. In the first submitted manuscript, we conduct a running average to examine the relationship between them, but since this may affect the time lag, we performed the 20-day lowpass filter in the manuscript submitted last time. Although the 20-day lowpass filter is effective in finding the lag between two time-series, some short-period variability does not match well, so it is judged that it is insufficient to prove their relationship. Therefore, we performed spectral analysis as shown in the

Quantitatively, consider normalized cross correlation.

figure below to find the dominant frequencies and their influence on the two time-series. As a result, the energy of frequencies less than 100 days in both time-series was very weak compared to the seasonal cycle, although the 45-day frequency was slightly strong in the meridional velocity. Therefore, we performed a 90-day low pass filter to remove frequencies of less than 100 days, and the result is reflected in figure 3e. The cross-correlation result is added to the supplementary Figure 8.

Figure. Spectral analysis of vertical averaged meltwater fraction and meridional velocity, respectively.

Reviewers' Comments:

Reviewer #2:

Remarks to the Author:

I found the manuscript improved, especially in terms of readability. I am still a bit unsure about the conclusions of the manuscript and below are two main comments (mostly following comments in previous reviews).

Main Comments

- Line 132-139: I am not following the analysis where you remove the seasonal cycle to compare southward flow and buoyancy forcing. If you remove the seasonal cycle, then you cannot study the seasonality anymore. Considering that your study focuses on the seasonality, this filter seems not appropriate. I do suggest to quantify the baroclinic variability of the southward flow and compare it with the measured changes (at all available depths). If the barotropic flow is dominant, then you can more safely assume that the seasonality of the meridional flow is driven by surface stress.
- Figure 4: 1 month lag between surface stress and meridional flow is not immediately clear to me. If the signal is barotropic, how can it take so long? Please add a few sentences to better explain this, including the relationship between lag and filtering. Please also make sure that the cross-correlation accounts for the changing degrees of freedom caused by lags/filtering.

Minor comments

- Line 29: remove "in Summer".
- Line 131: please add reference for the model, and spell it completely.
- Line 176: "deeper in the water column".
- Line 211: "subglacial" -> "basal"
- Line 239: "we identified".
- Line 260: "meltwater flux was negative".
- Line 269-272: Add that you are referring to DIS.

Reviewer #3:

Remarks to the Author:

Much improved! Only very MINOR copy-like edits. Rewritten passages (with exception of the Discussion paragraphs that contain the line-by-line edits below) are very clear and concise. Responses to my comments, and those of the other reviewers, were thorough and convincing.

Line 235: Need that hyphen in "two-years"?

Line 235: ", identified" Missing subject. Maybe "we identified"?

Line 239: "e.g." Needed?

Line 240: ", as well as further north in the deep trough of the presently studied system³⁶" Overly complicated, in my opinion, just use the name.

Line 242: ", although it is not believed to substantially influence the ice shelf melt processes there." I think the "belief" concerns only the present day, and not more broadly the future (or past).

Line 251: Extra comma.

Line 280: "atmospheric circulation in the AS is much more complex and diverse than in any other sea in Antarctica" – I'm not sure the cited works would stand behind such a statement.

Line 296: ", and we will be more confident about the need for further studies, e.g., ice-ocean coupled numerical model study considering the ice shelf." – grammar, particularly the last half.

Response to reviews of “Seasonal variability of ocean circulation near the Dotson Ice Shelf, Antarctica”

Manuscript NCOMMS-21-02666B

Response to Reviewers

We would like to thank the reviewers and editors for their time and effort in reviewing the paper "Seasonal Variability in Ocean Circulation Near the Dotson Ice Shelf, Antarctica," submitted for publication in Nature Communications. Detailed responses to the comments and explanations how the manuscript was changed in response to the comments are provided below. Line numbers refer to the new version of the manuscript. All changes are highlighted in the manuscript. See step-by-step answers to reviewers' comments below. Here, all line numbers refer to revised manuscript files.

Reviewer #2.		
ID	Comment Major	Response
1	Line 132-139: I am not following the analysis where you remove the seasonal cycle to compare southward flow and buoyancy forcing. If you remove the seasonal cycle, then you cannot study the seasonality anymore. Considering that your study focuses on the seasonality, this filter seems not appropriate. I do suggest to quantify the baroclinic variability of the southward flow and compare it with the measured changes (at all available depths). If the barotropic flow is dominant, then you can more safely assume that the seasonality of the meridional flow is driven by surface stress.	In this study, the calculated buoyancy flux has a seasonality, so we expected that it might affect the seasonality of the meridional velocity like the previous study in PIG (Webber et al., 2017). However, the buoyancy flux could not explain the variability of the meridional velocity shorter than the seasonal cycle, as shown in Supplementary Figure 3. The substantial seasonal variability in sea ice concentration probably affects both meridional velocities induced by OSSC and buoyancy flux. If there is a significant correlation between buoyancy flux and meridional velocity with their seasonality removed, it can be said that the ocean circulation in front of the DIS is affected by the buoyancy flux similarly to the PIG. However, as you pointed out, the comparative method that removes the seasonality in the meridional velocity, which is very important in this study, seems inadequate. We would like to further emphasize the importance of OSSC variability by quantifying and comparing the effects of buoyancy flux and OSSC on the baroclinic and barotropic components of meridional velocity, respectively, as you suggested. The depth-averaged meridional velocity calculated from 31 days running averaged velocities in 40 m intervals from 400 – 600 m shown the variability of $-11.17 - 0.35 \text{ cm s}^{-1}$, and the variability width was 11.52 cm s^{-1} (see the below figure). On the other hand, the variation range of meridional velocity in the 400 m depth with the depth-averaged removed was $0.67 - 3.28 \text{ cm s}^{-1}$, and the variability width was 2.61 cm s^{-1}. And the variability width decreased with depth to 480 m and increased again below 500 m to the bottom. In the 600 m, it has shown the variability of $-4.2 - -0.86 \text{ cm s}^{-1}$, and the variability range was 3.34 cm s^{-1} (see the below table). On the other hand, the variation range of meridional velocities with depth-averaged removed (400-600 m) in 680 m obtained for one year were $-14.35 - -8.68 \text{ cm s}^{-1}$, and the variability width

was 5.67 cm s^{-1} . Thus, the variability of the barotropic component induced by OSSC was 2 times greater than that of the baroclinic component near the bottom.

We modified some sentence as follows.

Edit Line 133-137: “The buoyancy flux, which affects the density structure of the water column, modulates the baroclinic component of the meridional velocity. However, the variation range of the baroclinic component (depth-averaged removed) in meridional velocity was 5.67 cm s^{-1} at 680 m depth, which was the largest in the entire layer, but almost half of the barotropic component of 11.52 cm s^{-1} .”

Figure. Barotropic component (depth-averaged, black) and baroclinic component (depth-averaged removed, color line) of meridional velocity.

Table. Variation ranges of baroclinic component of meridional velocity (depth-averaged removed) at each depth.

Depth (m)	400	440	480	520	560	600	640	680
Max (cm s ⁻¹)	3.28	2.28	1.14	0.49	-0.3	-0.86	-4.75	-8.68
Max (cm s ⁻¹)	0.67	0.42	0.18	-0.61	-2.39	-4.2	-8.89	-14.35
Dif. (cm s ⁻¹)	2.61	1.86	0.96	1.1	2.09	3.34	4.14	5.67

2 Figure 4: 1 month lag between surface stress and meridional flow is not immediately clear to me. If the signal is barotropic, how can it take so long? Please add a few sentences to better explain this, including the relationship between lag and filtering. Please also make sure that the cross-correlation accounts for the changing degrees of freedom caused by lags/filtering.

In the main text, the figure about the relationship between meridional flow and OSSC is Figure 2, so I will consider it as a comment on Figure 2, not Figure 4.

In this study, we tried to identify the cause of variability in the meridional flow, which is essential for heat supply to the DIS. In particular, the meridional flow is more dependent on the wind-driven barotropic component, and OSSC was calculated to quantify it. Cross spectral analysis between the calculated OSSC and the barotropic component of meridional velocity showed the strongest coherence at the 80-day frequency. In addition, significant coherence was found in frequencies of 31 and 22 days. Barotropic southward flow is governed by the east-west sea level gradient, which is associated with the accumulation of OSSCs. Positive OSSC

		accelerates the southward flow by increasing the gradient of the sea level in the east-west direction. In contrast, negative OSSC decreases the gradient of the sea level and slows the southward flow. If it is assumed that the OSSC shows only a single frequency of 80-day, the OSSC is decreased from 0 to 20 days, but cumulative OSSC increases, accelerating the barotropic southward flow due to increasing of the sea level gradient. And then, the maximum barotropic southward flow will appear at 20-day. Eventually, the barotropic southward flow will show a 20-day delay from OSSC variability, which is 1/4 of an 80-day frequency. In addition, although coherence of 80-day frequency was maximum in cross-spectral analysis, shorter frequencies of 31 and 22 days were also statistically significant, and these short frequencies could affect the delay barotropic southward flow. We modified and added the explanation of the cross-spectral analysis results as follows. Edit line 151: “.. frequency with other shorter frequencies (e.g., 31-day and 22-day)..” Edit line 153-154: “.. approximately a quarter of a combines of a dominant 80-day frequency and other shorter frequencies..” Edit line 155-158: “That is, positive OSSC along the eastern slope of the DIS increases the cumulative OSSC, increasing the horizontal gradient of the sea surface elevation and accelerating the barotropic southward flow. When the positive OSSC turns negative, the barotropic southward flow is maximum and then decelerates.” Also, the confidence interval in figure2 was recalculated considering the variation in degrees of freedom, and the confidence interval was modified as 99%. In addition to Figure 2, SF8 and SF10 with cross-correlation analysis were also modified.
	Minor	
3	Line 29: remove “in Summer”.	Done. We removed ‘in summer’. (see the edit line 29)
4	Line 131: please add reference for the model, and spell it completely.	Done. Model name and reference (Mazloff, M. R et al., 2010) were added. That sentence changed as follows. Edit line 130-131: “... data-assimilating Southern Ocean State Estimate (SOSE) ²⁷ .”
5	Line 176: “deeper in the water column”	Done. We changed ‘deeper depth’ to ‘deeper in the water column’ (See the edit line 181)
6	Line 211: “subglacial” - > ”basal”	Done We changed ‘subglacial’ to ‘basal’. (see the edit line 216)

7	Line 239: “we identified”.	Done We added ‘we’. Sentence was changed as follows Edit Line 244: “..., we identified a substantial inflow of warm and salty water near the seabed at the eastern flank of the deep trough that leads into the ice shelf cavity.”
8	Line 260: “meltwater flux was negative”.	Done. ‘showed’ change to ‘was’. (see edit line 265)
9	Line 269-272: Add that you are referring to DIS	The mentioned sentence indicates an important result of this study that the spatial distribution of wind and sea ice in front of the DIS affects the fluctuations of heat transport. Unfortunately, there were no observation data on velocity and properties using long-term mooring in front of the DIS. Kim et al 2016 showed that circulation was changed by wind and sea ice conditions in front of DIS, but it was not the result of analyzing heat transport. Kim et al., 2017, Dotto., 2020 also showed that the distribution of wind and sea ice had an effect on CDW inflow, but this result used long-term observation results from the Dotson trough. Unfortunately, it is difficult to cite existing papers in front of the DIS because the relationship between the variability of heat transport and sea ice and wind shown in this paper are results that have not been used in DIS so far.

a

Reviewer #3.		
ID	Comment Minor	Response
1	Line 235: Need that hyphen in “two-years”?	Done. We remove hyphen. (see the edit line 244)
2	Line 235: “, identified” Missing subject. Maybe “we identified”?	Done We added ‘we’. Sentence was changed as follows Edit Line 244: “..., we identified a substantial inflow of warm and salty water near the seabed at the eastern flank of the deep trough that leads into the ice shelf cavity.”
3	Line 239: “e.g.” Needed?	Done. We removed ‘e.g.’ (see the edit line 243)
4	Line 240: “, as well as further north in the deep trough of the presently studied system ³⁶ ” Overly	Done. That sentence changed as follows Edit Line 249: “..., as well as Dotson trough further north than the presently studied system ³⁷ .”

	complicated, in my opinion, just use the name.	
5	Line 242: “, although it is not believed to substantially influence the ice shelf melt processes there.” I think the “belief” concerns only the present day, and not more broadly the future (or past).	According to previous studies, it is known that the melting of ice shelves by warm water in the ocean is relatively small in 'cold' ice shelves such as ROSS and Filchner-Ronne. This result is a 'clear' fact known from the past to the present. Therefore, the ambiguous 'belief' was removed and the sentence was revised as follows. Edit Line 251 : “...,although it does not substantially influence the ice shelf melt processes there.”
6	Line 251: Extra comma.	Done. We added extra comma before ‘that’ as follows. Actually, I'm not sure if this is correct. Edit Line 259-261: “The substantial seasonal variability of heat transport and meltwater discharge near the ice shelf calving front, that was observed, suggests that previously evaluated heat- and meltwater transport^{15,24,25}, based on summer observation, may have been overestimated.”
7	Line 280: “atmospheric circulation in the AS is much more complex and diverse than in any other sea in Antarctica” – I’m not sure the cited works would stand behind such a statement.	Connolley et al., 2007 analyzed the mean sea level pressure using the climate model and found that the Amundsen sea had the greatest variability among the Antarctic Oceans. Citing this study, Turner et al., 2017 explained that “atmospheric circulation over the Amundsen Sea is more variable than any other region on Earth”. Therefore, I moved the location of the reference of Connolley (2007) and cite Turner (2017)’s paper. (see the edit line 290)
8	Line 296: “, and we will be more confident about the need for further studies, e.g., ice-ocean coupled numerical model study considering the ice shelf.” – grammar, particularly the last half.	Mentioned sentence “Furthermore, these results will improve our understanding of the effects of long-term variability in the atmosphere, such as climate change, on ice-shelf retreat, and we will be more confident about the need for further studies” has a similar meaning to the preceding sentence. (Therefore, understanding the long-term variability of the atmosphere and the subsequent response of the ocean circulation are likely essential for determining the long-term melting trend of the WAIS.) Some sentences have been deleted and modified as follows. Edit Line 304-305: “These results suggest that further studies are needed, e.g., ice-ocean coupled numerical model study considering the ice shelf.”

Reviewers' Comments:

Reviewer #2:

Remarks to the Author:

Well done! I think that now the manuscript is ready for publication. These new observations and results make an important contribution to our understanding of the Antarctic system.

I have one minor comment. Please write "Dotson Ice Shelf cavity" in the abstract (line 26).

Response to reviews of “Seasonal variability of ocean circulation near the Dotson Ice Shelf, Antarctica”

Manuscript NCOMMS-21-02666C

Response to Reviewers

We would like to thank the reviewers and editors for their time and effort in reviewing the paper "Seasonal Variability in Ocean Circulation Near the Dotson Ice Shelf, Antarctica," submitted for publication in Nature Communications. Detailed responses to the comments and explanations how the manuscript was changed in response to the comments are provided below. Line numbers refer to the new version of the manuscript. All changes are highlighted in the manuscript. See step-by-step answers to reviewers' comments below. Here, all line numbers refer to revised manuscript files.

Reviewer #2.		
ID	Comment	Response
1	Minor I have one minor comment. Please write "Dotson Ice Shelf cavity" in the abstract (line 26).	Done We added "cavity" after "Dotson ice shelf" (see the edit line 25).